# Boosting One-Point Derivative-Free Online Optimization via Residual Feedback

## Abstract

Zeroth-order optimization (ZO) typically relies on two-point feedback to estimate the unknown gradient of the objective function, which queries the objective function value twice at each time instant. However, if the objective function is time-varying, as in online optimization, two-point feedback can not be used. In this case, the gradient can be estimated using one-point feedback that queries a single function value at each time instant, although at the expense of producing gradient estimates with large variance. In this work, we propose a new one-point feedback method for online optimization that estimates the objective function gradient using the residual between two feedback points at consecutive time instants. We study the regret bound of ZO with residual feedback for both convex and nonconvex online optimization problems. Specifically, for both Lipschitz and smooth functions, we show that using residual feedback produces gradient estimates with much smaller variance compared to conventional one-point feedback methods, which improves the learning rate. Our regret bound for ZO with residual feedback is tighter than the existing regret bound for ZO with conventional one-point feedback and relies on weaker assumptions, which suggests that ZO with our proposed residual feedback can better track the optimizer of online optimization problems. We provide numerical experiments that demonstrate that ZO with residual feedback significantly outperforms existing one-point feedback methods in practice.

## 1 Introduction

Zeroth-order optimization (ZO) algorithms have been widely used to solve online optimization problems where first or second order information (i.e., gradient or Hessian information) is unavailable at each time instant. Such problems arise, e.g., in online learning and involve adversarial training Chen et al. (2017) and reinforcement learning Fazel et al. (2018); Malik et al. (2018) among others. The goal in online optimization is to minimize a sequence of time-varying objective functions $\{f_t(x)\}_{t=1:T}$, where the value $f_t(x_t)$ is revealed to the agent after an action $x_t$ is selected and is used to adapt the agent's future strategy. Since the future objective functions are not known *a priori*, the performance of the online decision process can be measured using notions of regret, generally defined as the difference between the total cost incurred by the decision selected by the agent online and the cost of the fixed or varying optimal decision that a clairvoyant agent could select.

Perhaps the most popular zeroth-order gradient estimator is the two-point estimator that has been extensively studied in Agarwal et al. (2010); Ghadimi & Lan (2013); Duchi et al. (2015); Ghadimi et al. (2016); Bach & Perchet (2016); Nesterov & Spokoiny (2017); Gao et al. (2018); Roy et al. (2019). Specifically, the two-point estimator queries the function value $f_t(x)$ for twice, for two different realizations of the decision variables, and uses the difference in these function values to estimate the desired gradient, as illustrated by the equation

$$\text{(Two-point feedback):} \quad \widetilde{g}_t^{(2)}(x) = \frac{u}{\delta}\Big(f_t(x + \delta u) - f_t(x)\Big), \tag{1}$$

where $\delta > 0$ is a parameter and $u \sim \mathcal{N}(0, I)$. However, the two-point gradient estimator can not be used for the solution of non-stationary online optimization problems that arise frequently, e.g., in online learning. The reason is that in these non-stationary online optimization problems, the objective

function being queried is time-varying, and hence only a single function value can be sampled at a given time instant. In this case, the following one-point feedback can be used

$$\text{(One-point feedback):} \quad \widetilde{g}_t^{(1)}(x) = \frac{u}{\delta} f_t(x + \delta u), \tag{2}$$

which queries the objective function $f_t(x)$ only once at each time instant. One-point feedback was first proposed and analyzed in Flaxman et al. (2005) for the solution of online convex optimization problems. Saha & Tewari (2011); Hazan & Levy (2014); Dekel et al. (2015) showed that the regret of convex online optimization methods using one-point gradient estimation can be improved assuming smoothness or strong convexity of the objective functions and using self-concordant regularization. More recently, Gasnikov et al. (2017) developed such regret bounds for stochastic convex problems. On the other hand, Hazan et al. (2016) characterized the convergence of one-point zeroth-order methods for static stochastic non-convex optimization problems. However, as shown in these studies, a limitation of one-point feedback is that the resulting gradient estimator has large variance and, therefore, induces large regret. In addition, the regret analysis for ZO with one-point feedback usually requires the strong assumption that the function value is uniformly upper bounded over time, so this method can not be used for practical non-stationary optimization problems.

**Contributions:** In this paper, we propose a novel one-point gradient estimator for zeroth-order online optimization and develop new regret bounds to study its performance. Specifically, our contributions are as follows. We propose a new one-point feedback scheme which requires a single function evaluation at each time instant. This feedback scheme estimates the gradient using the residual between two consecutive feedback points and we refer to it as residual feedback. We show that our residual feedback induces a smaller gradient estimation variance than the conventional one-point feedback scheme in Flaxman et al. (2005); Gasnikov et al. (2017). Furthermore, we provide regret bounds for online convex optimization with our proposed residual feedback estimator. Our analysis relies on a weaker assumption than the one needed in the case of the conventional one-point estimator, and our proposed regret bounds are tighter especially when the value of the objective function is large. In addition, we provide regret bounds for online non-convex optimization with residual feedback. Finally, we present numerical experiments that demonstrate that the proposed residual-feedback estimator significantly outperforms the conventional one-point method in its ability to track the time-varying optimizers of online learning problems. To the best of our knowledge, this is the first time a one-point zeroth-order method is theoretically studied for online non-convex optimization problems. It is also the first time that a one-point gradient estimator demonstrates comparable empirical performance to that of the two-point method. We note that two-point estimators can only be used to solve online non-stationary learning problems in simulations, where the system can be hard coded to be fixed during two queries of the objective function values at two different decision variables.

**Related work:** Zeroth-order methods have been used to solve many different types of optimization problems. For example, Balasubramanian & Ghadimi (2018) apply ZO to solve a set-constrained optimization problem where the projection onto the constraint set is non-trivial. Gorbunov et al. (2018); Ji et al. (2019) apply a variance-reduced technique and acceleration schemes to achieve better convergence speed in ZO. Wang et al. (2018) improve the dependence of the iteration complexity on the dimension of the problem under an additional sparsity assumption on the gradient of the objective function. And Hajinezhad & Zavlanos (2018); Tang & Li (2019) apply zeroth-order oracles to distributed optimization problems when only bandit feedbacks are available at each local agents. Our proposed residual feedback oracle can be used to solve such online optimization problems as well. Also related is work by Zhang et al. (2015) that considers non-convex online bandit optimization problems with a single query at each time step. However, this method employs the exploration and exploitation bandit learning framework and the proposed analysis is restricted to a special class of non-convex objective functions. Finally, Agarwal et al. (2011); Hazan & Li (2016); Bubeck et al. (2017) study online bandit algorithms using ellipsoid methods. In particular, these methods induce heavy computation per step and achieve regret bounds that have bad dependence on the problem dimension. As a comparison, our one-point method is computation light and achieves regret bounds that have better dependence on the problem dimension.

## 2 PRELIMINARIES AND RESIDUAL FEEDBACK

We first introduce the classes of Lipschitz and smooth functions.

**Definition 2.1** (Lipschitz functions). *The class of Lipschtiz-continuous functions $C^{0,0}$ satisfies: for any $f \in C^{0,0}$, $|f(x) - f(y)| \leq L_0 \|x - y\|$, $\forall x, y \in \mathbb{R}^d$, where $L_0 > 0$ is the Lipschitz parameter. The class of smooth functions $C^{1,1}$ satisfies: for any $f \in C^{1,1}$, $\|\nabla f(x) - \nabla f(y)\| \leq L_1 \|x - y\|$, $\forall x, y \in \mathbb{R}^d$, where $L_1 > 0$ is the smoothness parameter.*

In ZO, the objective is to estimate the first-order gradient of a function using zeroth-order oracles. Necessarily, we need to perturb the function around the current point along all the directions uniformly in order to estimate the gradient. This motivates us to consider the Gaussian-smoothed version of the function $f$ as introduced in Nesterov & Spokoiny (2017), $f_\delta(x) := \mathbb{E}_{u \sim \mathcal{N}(0,1)}[f(x + \delta u)]$, where the coordinates of the vector $u$ are i.i.d standard Gaussian random variables. The following bounds on the approximation error of the function $f_\delta(x)$ have been developed in Nesterov & Spokoiny (2017).

**Lemma 2.2.** *Consider a function $f$ and its smoothed version $f_\delta$. It holds that*

$$|f_\delta(x) - f(x)| \leq \begin{cases} \delta L_0 \sqrt{d}, & \text{if } f \in C^{0,0}, \\ \delta^2 L_1 d, & \text{if } f \in C^{1,1}, \end{cases} \quad \text{and } \|\nabla f_\delta(x) - \nabla f(x)\| \leq \delta L_1 (d + 3)^{3/2}, \text{ if } f \in C^{1,1}.$$

The smoothed function $f_\delta(x)$ satisfies the following amenable property Nesterov & Spokoiny (2017).

**Lemma 2.3.** *If $f \in C^{0,0}$ is $L_0$-Lipschitz, then $f_\delta \in C^{1,1}$ with Lipschitz constant $L_1 = \sqrt{d}\delta^{-1}L_0$.*

Consider the following online bandit optimization problem.

$$\min_{x \in \mathcal{X}} \sum_{t=0}^{T-1} f_t(x), \tag{P}$$

where $\mathcal{X} \subset \mathbb{R}^d$ is a convex set and $\{f_t\}_t$ is a random sequence of objective functions. In this setting, the objective functions $\{f_t\}_t$ are unknown *a priori* and their derivatives are unavailable. At time $t$, a new objective function $f_t$ is randomly generated independent of an agent's decisions, and then the agent queries the objective function value at certain perturbed points and use them to update the current policy parameters. The goal of the agent is to minimize a certain regret function.

Such an online setting often occurs in non-stationary learning scenarios where either the system is time-varying on its own or a single query of the function $f_t$ changes the system state (i.e., $f_t$ changes to $f_{t+1}$). In this non-stationary setting, the conventional two-point feedback scheme is known to be impractical as it requires to evaluate $f_t$ at two different points at the same time $t$. Instead, it is natural to use the one-point feedback scheme (2) in Gasnikov et al. (2017). However, the gradient estimate based on the above one-point feedback induces a large variance that leads to a large regret. In this paper, we focus on such an one-point derivative-free setting and propose the following novel one-point residual feedback scheme for estimating the gradient with reduced variance.

$$\text{(Residual feedback):} \quad \widetilde{g}_t(x_t) := \frac{u_t}{\delta} \big( f_t(x_t + \delta u_t) - f_{t-1}(x_{t-1} + \delta u_{t-1}) \big), \tag{3}$$

where $u_{t-1}, u_t \sim \mathcal{N}(0, I)$ are independent random vectors. To elaborate, the residual feedback in (3) queries $f_t$ at a single perturbed point $x_t + \delta u_t$, and then subtracts it by $f_{t-1}(x_{t-1} + \delta u_{t-1})$ obtained from the previous iteration. We name such a scheme as *one-point residual feedback*. Next, we explore some basic properties of the residual feedback. We first show that this estimator is an unbiased gradient estimate of the smoothed function $f_{\delta,t}$.

**Lemma 2.4.** *The residual feedback satisfies $\mathbb{E}\big[\widetilde{g}_t(x_t)\big] = \nabla f_{\delta,t}(x_t)$ for all $x_t \in \mathcal{X}$ and $t$.*

*Proof.* By the fact that $u_t$ has zero mean and is independent from $u_{t-1}$ and $x_{t-1}$. □

We consider the following ZO algorithm with residual feedback

$$\text{(ZO with residual feedback):} \quad x_{t+1} = \Pi_{\mathcal{X}} \big( x_t - \eta \widetilde{g}_t(x_t) \big), \tag{4}$$

where $\eta$ is the learning rate and $\Pi_{\mathcal{X}}$ is the projection operator onto the set $\mathcal{X}$. The update (4) can be implemented assuming that the objective function can be queried at points outside the feasible set $\mathcal{X}$, similar to the methods considered in Duchi et al. (2015); Bach & Perchet (2016); Gasnikov et al. (2017). Note that it is possible to modify the update (4) so that the iterates are guaranteed to be within the feasible set $\mathcal{X}$. This modification and related analysis can be found in Section H in the

supplementary material. The requirement that the objective function is evaluated at feasible points in derivative-free optimization algorithms has also been considered in Bubeck et al. (2017); Bilenne et al. (2020). Specifically, Bubeck et al. (2017) develop the so called ellipsoid method, which requires computation of an ellipsoid containing the optimizer at each time step. On the other hand, almost concurrently with this work, Bilenne et al. (2020) proposed a similar oracle as in (3) for a static convex optimization problem with specific objective and constraint functions. Next, we bound the second moment of the gradient estimate based on the residual feedback.

**Lemma 2.5** (Second moment). *Assume that $f_t \in C^{0,0}$ with Lipschitz constant $L_0$ for all time $t$. Then, under the ZO update rule in (4), the second moment of the residual feedback satisfies: for all $t$,*

$$\mathbb{E}[\|\widetilde{g}_t(x_t)\|^2] \leq \frac{4dL_0^2\eta^2}{\delta^2}\mathbb{E}[\|\widetilde{g}_{t-1}(x_{t-1})\|^2] + D_t, \tag{5}$$

*where $D_t := 16L_0^2(d+4)^2 + \frac{2d}{\delta^2}\mathbb{E}\big[\big(f_t(x_{t-1} + \delta u_{t-1}) - f_{t-1}(x_{t-1} + \delta u_{t-1})\big)^2\big].$*

The above lemma shows that the second moment of the residual feedback can be bounded by a perturbed contraction, provided that we choose $\eta$ and $\delta$ such that the contracting rate $\alpha = 4dL_0^2\eta^2\delta^{-2} < 1$. As we show later in the analysis, such a contraction property leads to a small variance of the residual feedback that helps reduce the regret of the online ZO algorithm.

# 3 ZO WITH RESIDUAL FEEDBACK FOR ONLINE CONVEX OPTIMIZATION

In this section, we consider the online bandit problem (P) where the sequence of functions $\{f_t\}_{t=0:T-1}$ are all convex. In particular, we are interested in analyzing the following static regret of the algorithm.

$$R_T := \mathbb{E}\Big[\sum_{t=0}^{T-1} f_t(x_t) - \min_{x \in \mathcal{X}}\sum_{t=0}^{T-1} f_t(x)\Big]. \tag{6}$$

We make the following assumption on the non-stationary of the online learning problem.

**Assumption 3.1** (Bounded variation). *There exists $V_f > 0$ such that for all $t$,*

$$\mathbb{E}\big[|f_t(x_{t-1} + \delta u_{t-1}) - f_{t-1}(x_{t-1} + \delta u_{t-1})|^2\big] \leq V_f^2, \tag{7}$$

*where the expectation is taken over $x_{t-1}$, the random vector $u_{t-1}$ and the random functions $f_{t-1}, f_t$.*

Intuitively, we assume the squared variation of the objective function between two consecutive time instants is uniformly bounded over time. We note that this assumption is much weaker than the uniformly bounded function value assumption, i.e., $\mathbb{E}\big[f_t(x)^2\big] \leq B^2, \forall t, x \in \mathcal{X}$, which is used in the analysis of ZO with the conventional one-point feedback Gasnikov et al. (2017). In particular, under Assumption 3.1, the perturbation term in Lemma 2.5 can be bounded as $D_t \leq 16L_0^2(d+4)^2 + 2dV_f^2\delta^{-2}$. Then, by telescoping the contraction inequality, we obtain the following bound for the second moment of the residual-feedback gradient estimate,

$$\mathbb{E}[\|\tilde{g}_t(x_t)\|^2] \leq \max\Big\{\mathbb{E}[\|\tilde{g}_0(x_0)\|^2], \frac{1}{1-\alpha}\Big(16L_0^2(d+4)^2 + \frac{2d}{\delta^2}V_f^2\Big)\Big\}. \tag{8}$$

In practice, $\delta$ is usually chosen to be sufficiently small, and the above bound is dominated by $\mathcal{O}\big(d\delta^{-2}V_f^2\big)$, which is much smaller than the second moment bound of the conventional one-point feedback $\mathcal{O}(d\delta^{-2}B^2)$ ($B^2$ is the uniform bound of the second moment of $f_t$ over time). For example, consider the time-varying objective functions, $f_0(x) = 1/2x^2$ and $f_t(x) = f_{t-1}(x) + n_t$, where $n_t$ is Gaussian noise with zero mean at time $t$. Then, it can be verified that Assumption 3.1 holds with a finite $V_f$ whereas the second moment of $f_t(x)$ is unbounded over time. This suggests that the variance of the residual feedback can be significantly smaller than that of the conventional one-point feedback.

Next, we first consider the case where the objective function $f_t$ is convex and Lipschitz. Based on the above characterization of the second moment of residual feedback, we obtain the following regret bound for ZO with residual feedback.

**Theorem 3.2** (Regret for Convex Lipschitz $f_t$). *Let Assumption 3.1 hold. Assume that $f_t \in C^{0,0}$ is convex with Lipschitz constant $L_0$ for all $t$ and $\|x_0 - x^*\| \leq R$. Run ZO with residual feedback for $T > R^2$ iterations with $\eta = R^{\frac{3}{2}}(2\sqrt{2}L_0\sqrt{d}T^{\frac{3}{4}})^{-1}$ and $\delta = \sqrt{R}T^{-\frac{1}{4}}$. Then, we have that*

$$R_T \leq \sqrt{2}L_0\sqrt{dR}T^{\frac{3}{4}} + \frac{\mathbb{E}\big[\|\tilde{g}_0(x_0)\|^2\big]R^{\frac{3}{2}}}{2\sqrt{2d}L_0T^{\frac{3}{4}}} + 8\sqrt{2}\frac{(d+4)^2}{\sqrt{d}}L_0R^{\frac{3}{2}}T^{\frac{1}{4}}$$
$$+ 2L_0\sqrt{dR}T^{\frac{3}{4}} + \sqrt{2dR}V_f^2L_0^{-1}T^{\frac{3}{4}}. \tag{9}$$

*Asymptotically, we have $R_T = \mathcal{O}((L_0 + L_0^{-1}V_f^2)\sqrt{dR}T^{\frac{3}{4}})$.*

To the best of our knowledge, the best known regret for ZO with the conventional one-point feedback is of the order $\mathcal{O}(\sqrt{dL_0RB}T^{\frac{3}{4}})$ Gasnikov et al. (2017). Therefore, our regret bound is tighter if the function variation satisfies $V_f^2 \leq \mathcal{O}(B^{\frac{1}{2}}L_0^{\frac{3}{2}})$. Essentially, using the proposed residual feedback gradient estimator, the regret of ZO no longer depends on the uniform bound of the function value, which can be huge in practice. Instead, our regret only relies on how fast the function varies over time.

**Remark 3.3.** *We note that the complexity bound in Theorem 3.2 generally depends on the values of the Lipschitz parameters $L_0$, $L_1$ and the constant $V_f^2$. Specifically, choose $\eta = R^{\frac{3}{2}}(2\sqrt{2}L_0\sqrt{d}T^{\frac{3}{4}})^{-1}$ and $\delta = \sqrt{R}L_0^{-q}T^{-\frac{1}{4}}$ with $q > 0$ as a tuning parameter, and we obtain that $R_T = \mathcal{O}((L_0 + L_0^{1-q} + L_0^{2q-1}V_f^2)\sqrt{dR}T^{\frac{3}{4}})$ when $T \geq L_0^{2q}R^2$. If $L_0 < 1$, we can choose $q = 1$ to achieve the bound $R_T = \mathcal{O}((L_0 + L_0V_f^2)\sqrt{dR}T^{\frac{3}{4}})$. On the other hand, if $L_0 \geq 1$, we can choose $q = 0$ to achieve the bound $R_T = \mathcal{O}((L_0 + L_0^{-1}V_f^2)\sqrt{dR}T^{\frac{3}{4}})$. We note that the dependence of the bounds in Theorems 3.4, 4.2 and 4.3 on $L_0, L_1$ can also be optimized in a similar way by properly choosing $\delta$.*

Next, we present the regret of ZO with residual feedback for convex smooth objective functions.

**Theorem 3.4** (Regret for Convex Smooth $f_t$). *Let Assumption 3.1 hold. Assume that $f_t(x) \in C^{0,0} \cap C^{1,1}$ is convex with Lipschitz constant $L_0$ and smoothness constant $L_1$ for all $t$, and assume that $\|x_0 - x^*\| \leq R$. Run ZO with residual feedback for $T > R^2$ iterations with $\eta = R^{\frac{4}{3}}(2\sqrt{2}L_0d^{\frac{2}{3}}T^{\frac{2}{3}})^{-1}$ and $\delta = R^{\frac{1}{3}}d^{-\frac{1}{6}}T^{-\frac{1}{6}}$. Then, we have that*

$$R_T \leq \sqrt{2}L_0d^{\frac{2}{3}}R^{\frac{2}{3}}T^{\frac{2}{3}} + \frac{\mathbb{E}\big[\|\tilde{g}_0(x_0)\|^2\big]R^{\frac{4}{3}}}{2\sqrt{2}L_0d^{\frac{2}{3}}T^{\frac{2}{3}}} + 8\sqrt{2}L_0\frac{(d+4)^2}{d^{\frac{2}{3}}}R^{\frac{4}{3}}T^{\frac{1}{3}}$$
$$+ 2L_1d^{\frac{2}{3}}R^{\frac{2}{3}}T^{\frac{2}{3}} + \sqrt{2}L_0^{-1}d^{\frac{2}{3}}R^{\frac{2}{3}}V_f^2T^{\frac{2}{3}}. \tag{10}$$

*Asymptotically, the above regret bound is in the order of $\mathcal{O}((L_0 + L_1 + L_0^{-1}V_f^2)(dRT)^{\frac{2}{3}})$.*

To the best of our knowledge, the best known regret for ZO with the conventional one-point feedback in the convex smooth case is of the order $\mathcal{O}(L_1^{\frac{1}{3}}(dRBT)^{\frac{2}{3}})$ Gasnikov et al. (2017). Therefore, our regret bound is tighter if the function variation satisfies $V_f^2 \leq \mathcal{O}(B^{\frac{2}{3}}L_1^{\frac{1}{3}}L_0)$. Our numerical experiments show that ZO with residual feedback always outperforms ZO with the conventional one-point feedback in practice.

## 4 ZO WITH RESIDUAL FEEDBACK FOR ONLINE NONCONVEX OPTIMIZATION

In this section, we analyze the regret of ZO with residual feedback in solving the unconstrained online bandit problem (P) with nonconvex functions. Throughout this section, we make the following assumption regarding the objective functions.

**Assumption 4.1.** *There exist $W_T, \widetilde{W}_T > 0$ such that the following conditions hold for all $t$.*

1. *$\sum_{t=1}^{T} \mathbb{E}[f_{\delta,t}(x_t) - f_{\delta,t-1}(x_t)] \leq W_T$, where the expectation is taken with respect to $x_t$ and the random smoothed objective functions $f_{\delta,t-1}$, $f_{\delta,t}$.*
2. *$\sum_{t=1}^{T} \mathbb{E}[|f_t(x_{t-1} + \delta u_{t-1}) - f_{t-1}(x_{t-1} + \delta u_{t-1})|^2] \leq \widetilde{W}_T$, where the expectation is taken with respect to $x_{t-1}$, the random vector $u_{t-1}$ and the random objective functions $f_{t-1}$, $f_t$.*

The above two conditions measure the accumulated first-order and second-order function variations, as also adopted by Roy et al. (2019).

Next, we consider the case where $\{f_t\}_t$ are nonconvex and Lipschitz continuous functions. Since the objective function $f_t$ is not necessarily differentiable, i.e., $\nabla f(t)$ is not well defined, we define the regret as the accumulated gradient of the smoothed function, i.e., $R_{g,\delta}^T := \sum_{t=0}^{T-1} \mathbb{E}[\|\nabla f_{\delta,t}(x_t)\|^2]$. In addition, it is often required that the smoothed function $f_{\delta,t}$ is close to the original function $f_t$ such that $|f_{\delta,t}(x) - f_t(x)| \leq \epsilon_f$ for all $t$. To satisfy this condition, we need to choose $\delta \leq (\sqrt{d}L_0)^{-1}\epsilon_f$ according to Lemma 2.2. We obtain the following regret bound for ZO with residual feedback.

**Theorem 4.2** (Nonconvex Lipschitz $f_t$). *Let Assumptions 4.1 hold. Assume that $f_t \in C^{0,0}$ with Lipschitz constant $L_0$ and that $f_t$ is bounded below by $f_t^*$ for all $t$. Run ZO with residual feedback for $T > (d\epsilon_f)^{-1}$ iterations with $\eta = \epsilon_f^{\frac{3}{2}}(2\sqrt{2}L_0^2 d^{\frac{3}{2}}T^{\frac{1}{2}})^{-1}$ and $\delta = \epsilon_f(d^{\frac{1}{2}}L_0)^{-1}$. Then, we have that*

$$
R_{g,\delta}^T \leq 2\sqrt{2}L_0^2\big(\mathbb{E}[f_{\delta,0}(x_0)] - f_{\delta,T}^* + W_T\big)d^{\frac{3}{2}}\epsilon_f^{-\frac{3}{2}}T^{\frac{1}{2}} + \frac{\epsilon_f^{\frac{1}{2}}\mathbb{E}\big[\|\tilde{g}_0(x_0)\|^2\big]}{2\sqrt{2dT}}
$$

$$
+ 4\sqrt{2}L_0\epsilon_f^{\frac{1}{2}}\frac{(d+4)^2}{d^{\frac{1}{2}}}T^{\frac{1}{2}} + \frac{L_0^2}{\sqrt{2}}\frac{d^{\frac{3}{2}}\widetilde{W}_T}{\epsilon_f^{\frac{3}{2}}T^{\frac{1}{2}}}. \tag{11}
$$

*Asymptotically, we have $R_{g,\delta}^T = \mathcal{O}(d^{\frac{3}{2}}L_0^2\epsilon_f^{-\frac{3}{2}}(W_T + \widetilde{W}_T T^{-1})T^{\frac{1}{2}} + d^{\frac{3}{2}}L_0\epsilon_f^{\frac{1}{2}}T^{\frac{1}{2}})$.*

Based on Theorem 4.2, we observe that the regret bound satisfies $R_{g,\delta}^T/T \to 0$ whenever $W_T = o(T^{\frac{1}{2}}\epsilon_f^{\frac{3}{2}})$ and $\widetilde{W}_T = o(T^{\frac{3}{2}}\epsilon_f^{\frac{3}{2}})$. In particular, if the bounded variation Assumption 3.1 holds, then we have $\widetilde{W}_T \leq \mathcal{O}(TV_f^2)$, and it suffices to let $T^{-\frac{1}{2}}\epsilon_f^{-\frac{3}{2}} = o(1)$.

Next, we consider the nonconvex and smooth problem and study the regret $R_g^T := \sum_{t=0}^{T-1}\mathbb{E}[\|\nabla f_t(x_t)\|^2]$. We obtain the following regret for ZO with residual-feedback.

**Theorem 4.3** (Nonconvex smooth $f_t$). *Let Assumptions 4.1 hold. Assume that $f_t \in C^{0,0} \cap C^{1,1}$ with Lipschitz constant $L_0$ and smoothness constant $L_1$ and that $f_t$ is bounded below by $f_t^*$ for all $t$. Run ZO with residual feedback for $T$ iterations with $\eta = (2\sqrt{2}L_0 d^{\frac{4}{3}}T^{\frac{1}{2}})^{-1}$ and $\delta = (d^{\frac{5}{6}}T^{\frac{1}{4}})^{-1}$. Then,*

$$
R_g^T \leq 4\sqrt{2}L_0\big(\mathbb{E}[f_{\delta,0}(x_0)] - f_{\delta,T}^* + W_T\big)d^{\frac{4}{3}}T^{\frac{1}{2}} + \frac{L_1\mathbb{E}\big[\|\tilde{g}_0(x_0)\|^2\big]}{\sqrt{2}L_0 d^{\frac{4}{3}}T^{\frac{1}{2}}}
$$

$$
+ 8\sqrt{2}L_1L_0\frac{(d+4)^2}{d^{\frac{4}{3}}}T^{\frac{1}{2}} + \frac{\sqrt{2}L_1}{L_0}d^{\frac{4}{3}}\widetilde{W}_T + 2L_1^2\frac{(d+3)^3}{d^{\frac{5}{3}}}T^{\frac{1}{2}}. \tag{12}
$$

*Asymptotically, the above regret bound is in the order of $\mathcal{O}(d^{\frac{4}{3}}L_0 W_T T^{\frac{1}{2}} + d^{\frac{4}{3}}L_1 L_0^{-1}\widetilde{W}_T)$.*

Based on Theorem 4.3, we observe that the regret bound satisfies $R_g^T/T \to 0$ whenever $W_T = o(T^{\frac{1}{2}})$ and $\widetilde{W}_T = o(T)$. We note that these requirements of $W_T, \widetilde{W}_T$ are more relaxed than those in the nonsmooth case, as they do not rely on the small parameter $\epsilon_f$.

## 5 ZO WITH RESIDUAL FEEDBACK FOR STOCHASTIC ONLINE OPTIMIZATION

In this section, we generalize the residual feedback to solve stochastic online bandit problems. Since its regret analysis follows the same proof logic as that of ZO with residual feedback, we only introduce the key technical lemmas and comment on the proof difference. The stochastic online bandit problems are formulated as follows.

$$
\min_{x \in \mathcal{X}} \sum_{t=0}^{T-1}\mathbb{E}[F_t(x;\xi_t)], \quad \text{where } \mathbb{E}[F_t(x;\xi_t)] = f_t(x), \forall t, \tag{R}
$$

where $\xi_t$ denotes a certain noise that is independent of $x$. Different from the previous deterministic online setting, the agent in the stochastic setting can only query noisy evaluations of the function.

This covers the scenarios where the agent does not have access to the underlying data distribution. To solve the above stochastic online problem, we propose the following stochastic residual feedback

$$\widetilde{g}_t(x_t) := \frac{u_t}{\delta}\big(F_t(x_t + \delta u_t; \xi_t) - F_{t-1}(x_{t-1} + \delta u_{t-1}; \xi_{t-1})\big), \tag{13}$$

where $\xi_{t-1}$ and $\xi_t$ are independent random samples that are sampled in the iterations $t-1$ and $t$, respectively. Since the noisy function value $F(x; \xi_t)$ is an unbiased estimate of the objective function $f_t(x)$, it is straightforward to show that (13) is an unbiased gradient estimate of the function $f_{\delta,t}(x)$.

To analyze the regret of ZO with stochastic residual feedback, we first consider the **convex setting** and make the following assumption that bounds the variation of the stochastic functions.

**Assumption 5.1.** *(Bounded stochastic variation) There exists $V_{f,\xi} > 0$ such that for all $t$,*

$$\mathbb{E}\big[\big(F_t(x_{t-1} + \delta u_{t-1}, \xi_t) - F_{t-1}(x_{t-1} + \delta u_{t-1}, \xi_{t-1})\big)^2\big] \leq V_{f,\xi}^2,$$

*where the expectation is taken with respect to $x_{t-1}$, the random vector $u_{t-1}$ and the random objective functions $F_{t-1}(\cdot, \xi_{t-1})$, $F_t(\cdot, \xi_t)$.*

The above assumption generalizes Assumption 3.1 to the stochastic setting. The bound $V_{f,\xi}^2$ controls both the variation of function over time and the variation due to stochastic sampling.

The following lemma characterizes the second moment of the stochastic residual feedback.

**Lemma 5.2.** *Assume $F(x, \xi) \in C^{0,0}$ with Lipschitz constant $L_0$ for all $\xi$. Then, under the ZO update rule, we have that*

$$\mathbb{E}[\|\widetilde{g}_t(x_t)\|^2] \leq \frac{4dL_0^2\eta^2}{\delta^2}\mathbb{E}[\|\widetilde{g}_t(x_{t-1})\|^2] + D_{t,\xi},$$

*where $D_{t,\xi} := 16L_0^2(d+4)^2 + \frac{2d}{\delta^2}\mathbb{E}[\big(F_t(x_{t-1} + \delta u_{t-1}, \xi_t) - F_{t-1}(x_{t-1} + \delta u_{t-1}, \xi_{t-1})\big)^2].$*

Observe that the above second moment bound is very similar to that in Lemma 2.5, and the only difference is the perturbation term. In particular, the perturbation term $D_{t,\xi}$ can be further bounded by leveraging Assumption 5.1, and the resulting second moment bound is almost the same as that in eq. (8) for the deterministic case (simply replace $V_f$ in eq. (8) by $V_{f,\xi}$). Therefore, the regret analysis of ZO with stochastic residual feedback is the same as that of ZO with residual feedback in the deterministic online setting. Consequently, ZO with stochastic residual feedback achieves almost the same regret bounds as those in Theorems 3.2 and 3.4, and one simply needs to replace $V_f$ by $V_{f,\xi}$.

For the **nonconvex setting**, we adopt the following assumption that generalizes Assumption 4.1.

**Assumption 5.3.** *There exists $W_T, \widetilde{W}_{T,\xi} > 0$ such that the following two conditions hold for all $t$.*

1. $\sum_{t=1}^T \mathbb{E}[f_{\delta,t}(x_t) - f_{\delta,t-1}(x_t)] \leq W_T$, *where the expectation is taken with respect to $x_t$ and the random smoothed objective functions $f_{\delta,t-1}$, $f_{\delta,t}$.*
2. $\sum_{t=1}^T \mathbb{E}[|F_t(x_{t-1} + \delta u_{t-1}; \xi_t) - F_{t-1}(x_{t-1} + \delta u_{t-1}; \xi_{t-1})|^2] \leq \widetilde{W}_{T,\xi}$, *where the expectation is taken with respect to $x_{t-1}$, the random vector $u_{t-1}$ and the random objective functions $F_{t-1}(\cdot, \xi_{t-1})$, $F_t(\cdot, \xi_t)$.*

Then, following the same proof logic as that of Theorems 4.2 and 4.3, on can obtain similar regret bounds for ZO with stochastic residual feedback (simply replace $W_T, \widetilde{W}_T$ in Theorems 4.2 and 4.3 by $W_{T,\xi}, \widetilde{W}_{T,\xi}$, respectively).

# 6  NUMERICAL EXPERIMENTS

In this section, we compare the performance of ZO with one-point, two-point and residual feedback in solving two non-stationary reinforcement learning problems, i.e., LQR control and resource allocation, in which either the reward or transition functions are varying over episodes.

## 6.1  NONSTATINOARY LQR CONTROL

We consider an LQR problem with noisy system dynamics. The static version of this problem is considered in Fazel et al. (2018); Malik et al. (2018). Specifically, consider a system whose state

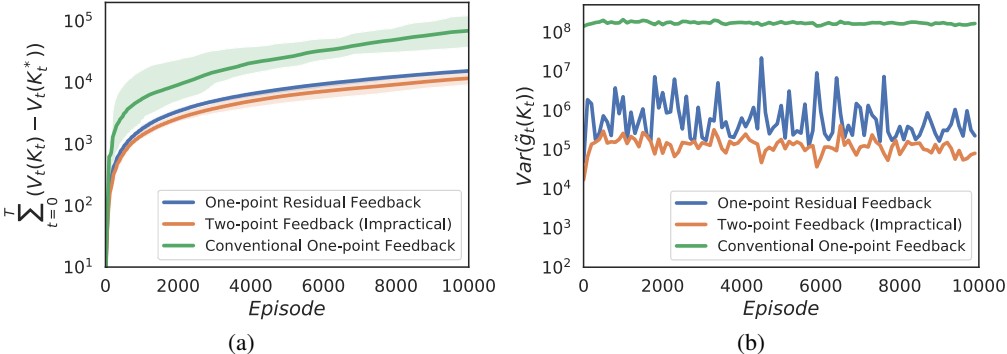

Figure 1: The regrets of applying the proposed residual one-point feedback (3) (blue), the two-point oracle in Bach & Perchet (2016) (orange) and the conventional one-point oracle in Gasnikov et al. (2017) (green) to online policy optimization for the nonstationary LQR problem. In (a), the regrets $\sum_{t=0}^{T} |V(K_t) - V(K^*)|$ of three methods are presented. In (b), the variance of the gradient estimates given by three methods are presented. The two point method (orange) is infeasible to use in practice and is presented here to serve as the simulating benchmark.

$x_k \in \mathbb{R}^{n_x}$ at step $k$ is subject to a transition function $x_{k+1} = A_t x_k + B_t u_k + w_k$, where $u_k \in \mathbb{R}^{n_u}$ is the action at step $k$, and $A_t \in \mathbb{R}^{n_x \times n_x}$ and $B_t \in \mathbb{R}^{n_x \times n_u}$ are dynamical matrices in episode $t$. These matrices are unknown and changing over episodes. The vector $w_k$ is the noise on the state transition. Specifically, the entries of the dynamical matrices $A_0$ and $B_0$ at episode 0 are randomly generated from a Gaussian distribution $\mathcal{N}(0, 0.1^2)$. Then, we generate the time-varying dynamical matrices as $A_{t+1} = A_t + 0.01 M_t$ and $B_{t+1} = B_t + 0.01 N_t$, where $M_t$ and $N_t$ are random matrices whose entries are uniformly sampled from [0,1]. Moreover, consider a state feedback policy $u_k = K_t x_k$, where $K_t \in \mathbb{R}^{n_u \times n_x}$ is the policy parameter that is fixed within episode $t$. Within each episode, there exists an optimal policy $K_t^*$ so that the discounted accumulated cost function $V_t(K) := \mathbb{E}\left[ \sum_{k=0}^{H-1} \gamma^k (x_k^T Q x_k + u_k^T R u_k) \right]$ at episode $t$ is minimized, where $\gamma \leq 1$ is the discount factor and $H$ is the horizon. The goal is to track the time-varying optimal policy parameter $K_t^*$ so that $V_t(K_t) - V_t(K_t^*)$ is small in every episode.

We apply the conventional one-point method in Gasnikov et al. (2017) and the proposed residual-feedback method (13) to solve the above non-stationary LQR problem. The performance of the two-point method in Bach & Perchet (2016) is also presented as a benchmark, although it is impractical in non-stationary scenarios. This is because the two-point method in Bach & Perchet (2016) requires to evaluate value function $V_t$ for two different policy functions at two consecutive episodes. However, evaluating the value function $V_t$ for a given policy during episode $t$ requires to collect samples by executing this policy. Then, during the subsequent episode $t+1$, since the problem is non-stationary, the dynamic matrices change to $A_{t+1}, B_{t+1}$ and so does the value function $V_{t+1}$. Therefore, it is not possible to evaluate the same value function $V_t$ at two different episodes and, as a result, the two-point method in Bach & Perchet (2016) is not applicable here. Each algorithm is run for 10 trials, and the stepsizes are optimized respectively. The accumulated regrets $\sum_{t=0}^{T-1} |V(K_t) - V(K^*)|$ of these algorithms are presented in Figure 1(a). We observe that the residual feedback method achieves a much lower regret than the conventional one-point method and has a comparable performance to that of the impractical two-point method. Moreover, we present in Figure 1(b) the estimated variance of the gradient estimates of these three methods at the policy iterates over episodes. It can be seen that the variance of our proposed residual-feedback is close to the impractical two-point feedback and is much smaller than that of the conventional one-point feedback. This observation justifies our theoretical characterization of the second moment of the residual feedback.

## 6.2 Nonstationary Resource Allocation

We consider a multi-stage resource allocation problem with time-varying sensitivity to the lack of resource supply. Specifically, 16 agents are located on a $4 \times 4$ grid. During episode $t$, at step $k$, agent $i$ stores $m_i(k)$ amount of resources and has a demand for resources in the amount of $d_i(k)$. Also, agent $i$ decides to send a fraction of resources $a_{ij}(k) \in [0, 1]$ to its neighbors

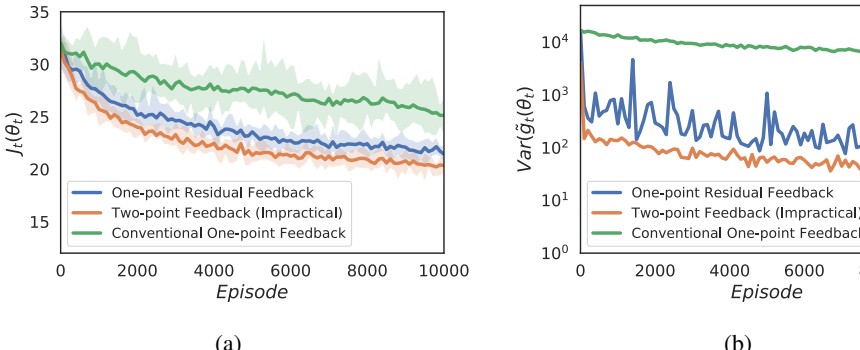

Figure 2: The costs during each episode by applying the proposed residual one-point feedback (3) (blue), the two-point oracle in Bach & Perchet (2016) (orange) and the conventional one-point oracle in Gasnikov et al. (2017) (green) to solve the non-stationary resource allocation problem are presented. In (a), the varying cost $J_t(\theta_t)$ of three methods are presented. In (b), the variance of the gradient estimates at agent 1 given by three methods are presented. The two point method (orange) is infeasible to use in practice and is presented here to serve as the simulating benchmark.

$j \in \mathcal{N}_i$ on the grid. The local amount of resources and demands of agent $i$ evolve as $m_i(k + 1) = m_i(k) - \sum_{j \in \mathcal{N}_i} a_{ij}(k)m_i(k) + \sum_{j \in \mathcal{N}_i} a_{ji}(k)m_j(k) - d_i(k)$ and $d_i(k) = \psi_i \sin(\omega_i k + \phi_i) + w_{i,k}$, where $w_{i,k}$ is the noise in the demand. At each step $k$, agent $i$ receives a local cost $r_{i,t}(k)$, such that $r_{i,t}(k) = 0$ when $m_i(k) \geq 0$ and $r_{i,t}(k) = \zeta_t m_i(k)^2$ when $m_i(k) < 0$, where $\zeta_t$ represents the varying sensitivity of the agents to the lack of supply during episode $t$. Let agent $i$ makes its decisions according to a parameterized policy function $\pi_{i,t}(o_i; \theta_{i,t}) : \mathcal{O}_i \to [0, 1]^{|\mathcal{N}_i|}$, where $\theta_{i,t}$ is the parameter of the policy function $\pi_{i,t}$ at episode $t$, $o_i \in \mathcal{O}_i$ denotes agent $i$'s local observation. Specifically, we let $o_i(k) = [m_i(k), d_i(k)]^T$. Our goal is to track the time-varying optimal policy so that the accumulated cost over the grid $J_t(\theta_t) = \sum_{i=1}^{16} \sum_{k=0}^{H} \gamma^k r_{i,t}(k)$ during each episode is maintained at a low level, where $\theta_t = [\ldots, \theta_{i,t}, \ldots]$ is the policy parameter, $H$ is the problem horizon at each episode, and $\gamma$ is the discount factor.

In Figure 2(a), we present the costs achieved during each episode $J_t(\theta_t)$ with 10 trials using ZO with the residual-feedback, one-point and the impractical two-point feedback. It can be seen that our proposed residual-feedback achieves a cost $J_t(\theta_t)$ that is as low as the cost achieved by the impractical two-point feedback in such a non-stationary environment. In particular, both residual and two-point feedback perform much better than the conventional one-point feedback. Moreover, Figure 2(b) compares the estimated variances of these feedback schemes, and one can observe that the variance of the residual feedback is comparable to that of the two-point feedback and is much smaller than that of the conventional one-point feedback.

## 7    CONCLUSION

In this paper, we proposed a residual one-point feedback oracle for zeroth-order online learning problems, which estimates the gradient of the time-varying objective function using a single query of the function value at each time instant. We showed that the regret bound of the proposed residual feedback estimator can be much lower than that of the conventional one-point method in online convex optimization setting. In addition, we studied the gradient size regret bound of the residual-feedback estimator when it is applied to the online non-convex optimization problems. Numerical experiments on two non-stationary reinforcement learning problems were conducted and the proposed residual-feedback estimator was shown to significantly outperform the conventional one-point method in non-stationary online learning problems.

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
