# OpenReview forum: "Boosting One-Point Derivative-Free Online Optimization via Residual Feedback"
_ICLR.cc/2021/Conference — Reject_

### Official Review · AnonReviewer3 · 2020-10-23
**The zeroth-order algorithm in constrained case is problematic**

**Rating:** 4
**Confidence:** 4

**Review:**

1. Paper contribution summary
    This paper proposes a new one-point zeroth-order gradient estimation method for online optimization. Comparing to previous methods in the same scenario, this paper's method is shown to have smaller variance, which can improve the learning rate in certain cases including classes of convex Lipschitz functions, convex smooth functions, non-convex Lipschitz functions, as well as non-convex smooth functions.

2. Strong and weak points of the paper
    1) Strong points: The paper's focus is on one-point zeroth-order gradient estimate, which is a more realistic setting in non-stationary online optimization problems as compared to most popular two-point estimator due to the queried function is time-varying. The new estimate method is based on residual feedback from previous time's perturbed objective value, which can help improve the regret order when function variation is small. It also extends the finding to online non-convex optimization problems with different regret definitions. And the proposed new one-point zeroth-order gradient estimation method is shown to have smaller variance and improved performance in the two numerical examples.

    2) Weak points: The proposed zeroth-order update rule for constrained convex case in Eq.(4) is problematic. For the constrained case, it uses a projection to make sure x_{t+1} is feasible. However, when doing gradient estimation, it actually uses the perturbed x_{t+1} + \delta u_{t+1}, which may violate the constraint, making this update rule not feasible sometimes. The claimed improvement for convex cases is only in the constant order instead of the order of T even under this problematic update rule. The regret metrics used in the two online non-convex cases: non-convex Lipschitz, non-convex smooth are different from each other, which is very weird.

3. My recommendation
    I would suggest a rejection due to the problematic update rule in Eq.(4) and the constant order improvement.

4. Supporting arguments for my recommendation.
    First, the problematic update rule as discussed above. Second, the only constant order improvement of regret in convex cases. Third, the weird different regret metrics used in non-convex cases.

5. Questions
    1) Can the author/s confirm if the update in Eq.(4) is problematic?
    2) Why using two different regret metric in non-convex problems?
    3) For the example used in the paragraph under Eq.(8) to show that the assumption 3.1 is weaker than previous works' uniform boundedness, i feel that the uniform boundedness assumption in prior works can be improved to have only bounded first or second order expectation. Then the example's statement won't hold anymore. Can the author/s make any comments on this?

---

> ### Author Response · Authors · 2020-11-21
> **Response to Reviewer 3**
>
> We thank the reviewer for providing valuable feedback that helps us improve the quality of this paper. Below is a point-to-point response to the reviewer’s concerns.
>
> Q1: Concerns on the correctness of the update in (4).
>
> A: We thank the reviewer for raising this question. We want to first emphasize that the feasibility of update (4) pointed out by the reviewer is not a concern for our method and the same update has been widely adopted in the existing literature, e.g., see the papers [A-E] given below. This literature allows the function to be evaluated outside the constraint set. For example, in the update (3) and gradient estimate (6) in [A] and in the update (6) in [B], the authors first obtain the next iterate by projecting onto the constraint set and then perturb this new iterate and evaluate its function value to estimate the zeroth-order gradient. Our update (4) adopts the same idea.
>
> On the other hand, we also want to mention that with minor modifications we can guarantee the feasibility of the perturbed points. We elaborate on these modifications in Section H in the supplementary material. In short, we can use unit sphere sampling instead of Gaussian sampling, and perform the projection onto a shrinked constraint set $(1-\xi)\mathcal{X}$ for some small $\xi>0$ (see eq.(48)), where the shrinkage coefficient $(1-\xi)$ is to make sure that the perturbed point is within the original set $\mathcal{X}$.
>
> To summarize, the update (4) is not problematic. To satisfy the additional requirement that the objective function can only be queried at feasible points, we’ve modified the algorithm so that the iterates can be guaranteed to lie within the feasible set and the related analysis is also provided. This discussion is added to the revised manuscript under (4).
>
> [A] Duchi et.al. (2015). Optimal rates for zero-order convex optimization: The power of two function evaluations.
>
> [B] Bach, F., & Perchet, V. (2016, June). Highly-smooth zero-th order online optimization.
>
> [C] Liu et.al. (2018, November). Zeroth-order stochastic projected gradient descent for nonconvex optimization.
>
> [D] Balasubramanian, K., & Ghadimi, S. (2018). Zeroth-order nonconvex stochastic optimization: Handling constraints, high-dimensionality and saddle-points.
>
> [E] Sahu, A. K., & Kar, S. (2020). Decentralized Zeroth-Order Constrained Stochastic Optimization Algorithms: Frank-Wolfe and Variants With Applications to Black-Box Adversarial Attacks.
>
> Q2: Explain the reason for using two different regret metrics in non-convex problems.
>
> A: Thanks for raising this question. We want to point out that these are standard metrics used for non-smooth & non-convex zeroth-order optimization, see Section 7 in the seminal work [Nesterov & Spokoiny, 2017]. To elaborate, in non-smooth non-convex optimization, the gradient of the objective function $\nabla f_t$ does not exist and hence the regret $R^T_g$ is not well-defined. In this case, we define a smoothed regret $R^T_{g,\delta}$ based on the smoothed objective function $f_{\delta, t}$. At the same time, we should not smooth the function too much, as otherwise $f_{\delta, t}$ can be very different from $f_t$. Thus, we control the smoothing parameter $\delta$ to enforce that $|f_{\delta, t} - f_t| \le \epsilon_f$. This discussion is presented above Theorem 4.2.
>
> Q3: I feel that the uniform boundedness assumption in prior works can be improved to have only bounded first or second order expectation.
>
> A: We agree that the one-point method can be analyzed using these relaxed conditions. For example, in the following paper [F], the analysis of the one-point method is based on bounded second moment of the objective function, i.e., $E[|f_t(x; \xi_t)|^2] \le B$. However, we note that our assumption only requires the second moment of the function variation to be bounded, which is less strict than the assumption that the objective function has bounded second moment. As an example, consider the time-varying functions defined by $f_0 = 1/2x^2$ and $f_t = f_{t-1} + n_t$, where $n_t\sim \mathcal{N}(0,1)$. Then, the second moment of $f_t$ increases linearly over time and is not bounded, but the variation $f_t - f_{t-1}$ has bounded second moment. We have included this example in the revised manuscript.
>
> [F] Gasnikov et.al. (2017). Stochastic online optimization. Single-point and multi-point non-linear multi-armed bandits. Convex and strongly-convex case.
>
> Q4: Constant order improvement.
>
> A: The constant-order improvement is expected, as our method is a one-point method and must not exceed the information theoretic limit of one-point methods. However, we think our one-point residual feedback has its own merit. It provides a more effective zero-order algorithm for solving online problems where two-point methods are not applicable, and resolves the large variance issue of the conventional one-point feedback.

---

> > ### Comment · AnonReviewer3 · 2020-11-22
> > **Feedback after author/s' rebuttal**
> >
> > I really appreciate the author/s' detailed feedback.
> > 1. I agree that the online convex update in Eq.(4) can be modified to satisfy the feasibility like shown in the updated appendix. But will that feasible update also have the same theorem results for the non-convex problems in Section 4? I really doubt it.
> > 2. The author/s argued that the reason to have two different metrics in online non-convex case is because of the non-smoothness. This is not satisfactory. For convex problem, it also has this non-smoothness, but it uses the sub-gradient instead. For the stochastic non-convex optimization, many of the convergence results are at least based on gradient or sub-gradient instead of the author/s' defined one, which is not rigorous in my opinion.
> > 3. The improvement in the regret is only constant in a certain range as shown below Theorem 3.2 and 3.3, which is very incremental. Also, there is no experiment to show the empirical performance w.r.t. the standard one-point update.
> > As a result, I will keep my score unchanged.

---

> > > ### Author Response · Authors · 2020-11-23
> > > **Response to Reviewer 3's feedback**
> > >
> > > We thank the reviewer for the comments. Please see our point-to-point response below:
> > >
> > > 1. We want to stress again that evaluation of function values outside the feasible set is a widely adopted setting considered in the zeroth order optimization literature; see, e.g., the methods in [A-E] that we have pointed out in our last response. Our contribution is to propose and study a novel one-point zeroth-order gradient estimator that can be used to solve practical online non-stationary optimization problems much more efficiently. This problem cannot be solved using two-point methods. While it can be solved using conventional one-point methods, these methods are known to have large gradient estimation variance.
> > >
> > > The concern raised by the reviewer, that the function values need to be queried within the feasibility set, can be important for specific applications. In our revised manuscript, we have proposed a solution to address this additional concern. However, we believe that extending this idea to more general settings is out of the main scope of this paper and should be addressed in another standalone paper. It’s impossible to address all the remaining questions in the field of zeroth-order optimization in one paper.
> > >
> > > 2. We do not understand why the reviewer thinks that the regret measure $R_{\delta, g}^T$ is not rigorous. This regret measure is adopted from Section 7 in [Nesterov & Spokoiny, 2017], and has a clear mathematical meaning, as we have discussed above Theorem 4.2. For convex problems, due to convexity, it is possible to directly measure the regret using the objective function value, as in (6). Therefore, a discussion on whether to use the gradient or subgradient to define the regret metrics in convex problems is not relevant.  For non-convex problems, we assume the reviewer refers to the optimization criterion for static optimization problems, where the goal is to find an optimal point $x^\ast$ so that $0 \in \partial f(x^\ast)$, where $ \partial f(x)$ denotes the subdifferential at $x$. However, such a subgradient may not even exist for general nonconvex and non-smooth functions. Even if a subgradient exists, it is not clear how to extend this static metric to a zeroth-order online case. To see this, note that a subgradient-based regret measure defined as $\sum_t ||\partial f_t(x)||^2$ is not well defined since $\partial f_t(x)$ is a set of vectors instead of one vector. In fact, the correct regret measure for general non-convex non-smooth non-stationary optimization problems is rather under-explored, especially in the gradient-based and corresponding zeroth-order optimization literature. The regret measure studied in this paper is in fact one of few criterions that can be used in this setting, which has clear mathematical meaning.
> > >
> > >
> > > 3. We thank the reviewer for the comments. Regarding the constant-wise improvement concern, we refer the reviewer to the study below:
> > >
> > > [H] Hu, X., Prashanth, L. A., György, A., & Szepesvari, C. (2016, May). (Bandit) convex optimization with biased noisy gradient oracles. In Artificial Intelligence and Statistics (pp. 819-828). PMLR.
> > >
> > > In this work, the authors have studied the optimal regret rate for gradient estimate oracles that have satisfied the property (5) in our paper for online convex problems. According to [H], a similar property to (5) is also satisfied by the conventional one-point oracle and two-point method oracle with uncontrolled noise samples. This suggests that it is theoretically impossible to achieve an order-wise improvement in the regret rate in these cases. However, this doesn’t mean that research on zeroth-order methods should stop with the conventional one-point method. This is the same case as with gradient-based methods whose optimal convergence rate has been analyzed years ago, but research did not end there and many more algorithms have been proposed because they provide better practical performance than earlier approaches. For example, ADAM does not supersede the optimal convergence rate of early gradient-based algorithms, but is still appreciated and widely used due to its nice empirical performance. Therefore, we think it is valuable to propose a different one-point zeroth-order oracle design, which has the same order of regret rate but performs much better in practice than the existing one-point zeroth-order method in non-stationary optimization problems.

---

> > > > ### Comment · AnonReviewer3 · 2020-11-24
> > > > **About author/s' additional feedback**
> > > >
> > > > Thanks again for the detailed feedback. I agree that it's unfair to request this paper to have a complete coverage on the non-convex case with feasible update. But my point is that the initial submission ignored such a critical difference and was not mentioned at all. When the author/s do consider the feasible update, it's not straightforward to have the similar results for non-convex cases as the convex case's.
> > > >
> > > > I also went through the discussion under Reviewer #1's comment. I agree with Reviewer #1 that assumption that f_t are random and independent is really rare for the online setting comparing to the adversarial case.
> > > >
> > > > Based on the whole discussions with all reviewers, I think it deserves a bit higher score. So I raised my score.

---

> > > > > ### Author Response · Authors · 2020-11-24
> > > > > **Response to Reviewer 3's feedback**
> > > > >
> > > > > Thanks for your feedback and we appreciate the gesture.
> > > > >
> > > > > We agree with the reviewer that querying the function at feasible points can be an important concern in some applications and we have discussed this issue under (4) in our previous revisions.
> > > > >
> > > > > On how to extend our results to the adversarial online optimization problems, please refer to point 1 in our updated reply to Reviewer 1 and Section i in the updated Supplementary material.

---

### Official Review · AnonReviewer1 · 2020-10-27
**A good paper with new theoretical results and empirical evidence**

**Rating:** 8
**Confidence:** 4

**Review:**

Summary:
The paper considers online optimization with zero-order oracle. Motivated by nonstationarity of the objective function, impracticality is underlined for the two-point feedback approach. Instead, staying in the one-point setting, the proposed approach reuses the objective value from the previous round of observations, which is called as residual feedback. The variance of the corresponding proxy for the subgradient is estimated under more relaxed assumptions than existing in the literature. The proposed approach leads to smaller variance and better regret bounds. Regret bounds are proved for smooth/non-smooth convex/non-convex cases, the non-convex case being analyzed for the first time in the literature. Numerical experiments show that the practical performance of the proposed gradient estimator is better than that of the existing one-point feedback methods and is close to the performance of the one-point approach with two observations per round. The latter approach can be impractical for some applications.

Evaluation:
I believe that the paper contains new interesting results on zero-order methods with one-point feedback, which are supported both theoretically and numerically. So, I suggest accepting the paper.

Pros:
1. New theoretical results which are significant for optimization and learning literature, as well as for applications.
2. Numerical results support theoretical findings.
3. The paper is overall clearly written and motivated.

Cons:
1. There are several minor comments mainly on the clarity of presentation. See below.


Minor comments
1. Some related work seems to be missing
http://proceedings.mlr.press/v48/hazanb16.pdf (non-convex optimization with one-point feedback)
https://link.springer.com/article/10.1007%2Fs10107-014-0846-1 (non-convex stochastic optimization)
http://papers.nips.cc/paper/5377-bandit-convex-optimization-towards-tight-bounds.pdf
http://papers.nips.cc/paper/4475-stochastic-convex-optimization-with-bandit-feedback.pdf
2. Please consider writing explicitly on p.7 that Bach & Perchet (2016) use two function evaluations in each round. Also it would be nice to explain in more details, why their approach is impractical. For example, in the considered in Sect. 6.1 example, why one can not observe x_{k+1} two times (with different values w_k), and the evaluate the loss twice?
3. The proof of Lemma 2.5 does not completely correspond to the statement of the Lemma. In the proof more is derived than stated in the Lemma, but under additional assumptions.
4. In the first line of Appendix F, did you mean that $f_{\delta,t} \in C^{1,1}$? Also here Assumption 3.1 is used, which should be mentioned.

---

> ### Author Response · Authors · 2020-11-21
> **Response to Reviewer 1**
>
> We thank the reviewer for providing valuable feedback that helps us improve the quality of this paper. Below is a point-to-point response to the reviewer’s concerns.
>
> Q1: Related works missing.
>
> A: We thank the reviewer for referring us to these works. These works study the two-point method, conventional one-point method or the ellipsoid method in static convex or non-convex optimization problems. We agree that they are also related and will be added to our discussion.
>
> Q2: Clarification on why the two-point method is impractical.
>
> A: Thanks for this comment. We will add the comment on [Bach & Perchet , 2016] according to the reviewer’s suggestion. If one wants to use the two-point gradient estimator (7) in [Bach & Perchet , 2016] to solve the non-stationary RL problem in Section 6.1, at episode $t$ with dynamical matrices $(A_t, B_t)$, (7) requires to evaluate two different policy parameters $K_t + \delta U_t$ and $K_t – \delta U_t$. However, in a practical non-stationary system, evaluation of each policy parameter requires one episode. For example, if we evaluate $K_t + \delta U_t$ at episode $t$, then the non-stationary environment evolves to episode $t+1$ with new dynamical matrices $(A_{t+1}, B_{t+1})$. Therefore, $K_t – \delta U_t$ cannot be evaluated in the environment with $(A_t, B_t)$ anymore. This violates the assumption in (7) in [Bach & Perchet , 2016]. Therefore, the two-point method (7) in [Bach & Perchet , 2016] cannot be implemented in a practical non-stationary system.
>
> Q3: Proof of Lemma 2.5.
>
> A: Thanks for this comment. Without Assumption 3.1, we can directly replace all $V_f^2$ in inequalities after (15) with $E[(f_t(z) – f_{t-1}(z))^2]$, where $z = x_{t-1} + \delta u_{t-1}$, and get the results in Lemma 2.5. This issue has been fixed in our revised supplementary material.
>
> Q4: Clarification on the proof in Appendix F
>
> A: Thanks for this comment. In the first line of Appendix F, we meant that when $f_t \in C^{1,1}$ with Lipschitz constant $L_1$, we have that $f_{\delta,t} \in C^{1,1}$ also with constant $L_1$. And the reviewer is correct that to get inequality (41), we also use Assumption 3.1. We have clarified this in the revised draft.

---

> > ### Comment · AnonReviewer1 · 2020-11-22
> > **Thanks for the answers, but I'm still confused and tend to lower the score**
> >
> > I would like to thank the authors for their answers. I've read the comments of the other reviewers and the answers. Several points still remain unclear to me.
> >
> > 1. It seems that there is a problem with the proof of Lemma 2.5. The problem is in the transition made in the last two lines of p.12 of the new supplementary: when taking the expectation, the argument inside $f_t$ depends on $u_t$ and there is also a multiplier with squared norm of $u_t$.  Since $f_t$ is also random, it seems that the inequality may not be correct due to interdependence of the different randomnesses. Moreover, it seems that at this point the inequality $E (X Y) \leq (EX)(EY)$ is used, which also may not hold.
> > It seems that the same problem affects Lemma 5.2.
> >
> > 2. The new assumptions 3.1, 4.1, 5.1 are in a sense confusing.
> >
> > a. Assumption 3.1. It is non-standard that the functions $f_t$ are now considered to be random. This assumption is not made when the problem statement is listed. Further, it seems that this assumption does not allow  to prove the central Lemma 2.5. The problem is in the transition made in the last two lines of p.12 of the new supplementary: when taking the expectation the argument inside $f_t$ depends on $u_t$ and there is also a multiplier with squared norm of $u_t$.  Since $f_t$ is also random, it seems that the inequality may not be correct due to interdependence of the different randomnesses. Moreover, it seems that at this point the inequality $E (X Y) \leq (EX)(EY)$ is used, which also may not hold.
> > I believe that an easy fix to prove Theorem 3.2 and 3.4 is to use Assumption 3.1 from the original submitted version of the paper, yet without any expectation.
> >
> > b. Assumption 4.1.1. I believe that the expectation w.r.t. all the sequence $x_t$ is sufficient here. At the same time the proof of Theorem 4.2 is quite confusing since it is not specified expectations w.r.t. what are taken. For example, in (33) the expectation is w.r.t. $u_t$, whereas in (34) it is already the full expectation. Moreover when making transition from (33) to (34) it is not mentioned that conditional expectations are iteratively applied to obtain full expectations. (Similar problems with other proofs). At the same time it should be explicitly pointed in the statement of Theorems 4.2, 4.3 that assumption 3.1 is made since it is used in the proof in the supplementary.
> >
> > c. Assumption 4.1.2. It seems that the proof of Theorem 4.2 under this assumption  is not completely correct for the reason stated in item 1 above. Here I don't see how this can be fixed.
> >
> > 3. Is seems that the same issues are presented in Sect. 5 and it would be nice to have the full proofs. What is especially confusing is that $V_f$ now depends on $\xi$. The reason is that in the derivations one has to take expectation w.r.t. $\xi$ and there is no assumption that $V_{f,\xi}^2$ is finite in expectation. Thus, I would not agree that everything is obtained by just replacing $V_f$ by $V_{f,\xi}$.
> >
> > 4. I'm confused by the LQR example. My understanding is the following. At the beginning of the episode $t$, the nature reveals to us the matrices $A_t,B_t$. We apply control given by $K_t$ and make $H$ steps of dynamics parameterized by iterations $k$. Why can't we have two sequences of $x_k$'s in parallel at the episode $t$ by applying two perturbed controls? Then, at  the end of the episode, we have two evaluations of $V_t$, yet with different noises given by realizations $w_k$. It seems that this is covered by [Bach & Perchet , 2016]'s setting. Moreover, if we don't know $A_t,B_t$, but know that they are fixed for $H$ steps, we can still run two trajectories in parallel by applying two controls.
> >
> > 5. I can't understand why "The function evaluation noise at time in [Bach & Perchet, 2016] is assumed to be zero-mean. As a comparison, our Assumptions 3.1, 4.1, 5.1 and 5.3 do not require the noise to be zero-mean, and instead we bound the time variation of the objective function." in the response to Reviewer 2. It seems that zero-mean assumption of  [Bach & Perchet, 2016] is the same assumption that is made in Sect. 5: $E F_t(x,\xi)=f_t(x)$.
> >
> >
> > To sum up, it seems that I was too optimistic in my first evaluation. Hoping that I have misunderstood something I lower my score to 6. At the same time, the bounds for the non-convex case, which I consider as the main contribution, are now questionable. So, I see a potential for further lowering the score.

---

> > > ### Author Response · Authors · 2020-11-23
> > > **(Cont'd) Response to Reviewer 1's feedback**
> > >
> > > 5. In both [Bach & Perchet, 2016] and our paper, we assume that the evaluation of the objective function is unbiased. The difference is in the noise model in the gradient estimator. In [Bach & Perchet, 2016], the gradient estimator (7) assumes that counterfactual function evaluation is possible, as we discussed in point 4. Then, the noise $\epsilon_n$ is zero mean. In our paper, we consider a more practical scenario, where counterfactual evaluation is not possible. As a result, although replacing $f_{t-1}$ with $f_t$ and adding the noise $\epsilon_n$ to the bracket in (3) we can obtain an estimator that is comparable to the estimator (7) in [Bach & Perchet, 2016], the noise $\epsilon_n$ in our case needs characterize the change from $f_{t-1}$ to $f_t$ and will not have zero mean. We describe our assumption on this noise term $\epsilon_n$ in Assumptions 3.1, 4.1, 5.1 and 5.3.
> > >
> > >
> > > 6. The bounds on the online non-convex case are not questionable. We believe the reviewer is referring to the questions raised by Reviewer 3, which do not question the correctness of the bounds, but the regret metrics used in online non-convex problems. To the best of our understanding, Reviewer 3 is asking why we do not directly extend the optimality criterion used in static non-convex non-smooth optimization problems to the online non-stationary case. We have clarified this question in point 2 in our new response to Reviewer 3.
> > >
> > > In addition, we would like to mention that our main contribution is to propose and study a novel one-point zeroth-order gradient estimator that can be used to solve a practical online non-stationary optimization problem much more efficiently, which cannot be solved by two-point methods and can be solved with conventional one-point method but with large gradient estimation variance. The regret measure we proposed to use in the online non-stationary non-convex non-smooth setting can be seen as a bonus point.

---

> > > > ### Comment · AnonReviewer1 · 2020-11-23
> > > > **Thank you for clarifications, I'm raising the score**
> > > >
> > > > Thank you very much for the provided clarifications. I now see that the proofs are correct.
> > > >
> > > > Despite I'm satisfied by the answers, I'm afraid that I can't totally agree with some points in the answers.
> > > >
> > > > 1. In my opinion, assumption that $f_t$ are random and independent is not common for the online setting, where the typical situation is that $f_t$ is adversarial to the actions of the optimizer.
> > > >
> > > > 2. Concerning $V_{f,\xi}$, the picture is still not completely clear to me. It seems that in the l.h.s. in the inequality of Assumption 5.1 the expectation w.r.t. $\xi$ is not taken, which leads to the conclusion that the l.h.s. depends on $\xi$ and it is not clear how to understand this inequality if the r.h.s. is a constant.
> > > >
> > > > 3. Concerning the LQR. I think it is too strong to say that "This is only possible in simulated environments where the user has full control of the system. In practice, this can never happen." For example we would like to learn a control for a drone to just fly from point A to point B. Then we can run two experiments with slightly different controls and apply the two-point estimator. (Yet, in this practical situations actually $A_t,B_t$ do not depend on $t$). Similarly we can run in parallel two chess games with slightly different controls and different players.
> > > >
> > > > 4. Concerning item 5 (comparison with [Bach & Perchet, 2016]). I see what the authors mean, but maybe I would not call it "noise" in the setting of this paper. My understanding is that noise is something uncontrolled, presented in the problem. On the opposite, the gradient estimator is something we construct. Maybe a word bias is more suitable here.
> > > >
> > > > It would be much appreciated if the other parts of the proofs would be also clarified by writing explicitly the expectation w.r.t. what is taken in each step. Like it was done for the transition between (33) and (34).
> > > >
> > > > To sum up, I'm pretty much satisfied and increase my score.

---

> > > > > ### Author Response · Authors · 2020-11-24
> > > > > **Response to Reviewer 1's feedback**
> > > > >
> > > > > We sincerely appreciate your very constructive feedback. Please see our point-to-point response below:
> > > > >
> > > > > 1. We agree with the reviewer that assuming the sequence {$f_t$} to be generated independently does not suit the adversarial scenario. The current results are presented for the case where the non-stationarity in the environment is caused by nature, rather than an adversarial agent.
> > > > >
> > > > > To account for the adversarial setting, we need to put some restrictions on what the adversary agent can do. For your interest, in Section i of the Supplementary material, we have included a discussion on the assumptions on the capability of the adversarial agent, which make the results presented in the paper still hold. How to relax these restrictions on the adversary agent will be an interesting question for adversarial online learning applications. We’ll definitely put it in our future research list.
> > > > >
> > > > > 2. We apologize for the confusion. In Assumption 5.1, on the l.h.s. of the inequality, the randomness of the evaluation noises $\xi_t$ is part of the randomness of the random functions $F_t(\cdot, \xi_t)$. Specifically, each random function $f_t$ and a random evaluation noise $\xi_t$ constitutes a random function $F_t(\cdot, \xi_t)$. Therefore, when we wrote that the expectation in Assumption 5.1 is taken over the random functions $F_{t-1}(\cdot, \xi_{t-1})$ and $F_t(\cdot, \xi_t)$, we meant that the expectation is also taken over both the random functions $f_t$, $f_{t-1}$ and the random evaluation noises $\xi_t$ and $\xi_{t-1}$. We hope that this is now more clear.
> > > > >
> > > > > 3. Thanks for your comments. We agree with the reviewer that in practical environments, if people know that the environment is not varying over time, as the example described by the reviewer, where the drone’s dynamics do not vary over different trials of experiments, two-point method can be implemented and its analysis still applies. In this paper, we study the non-stationary environment setting, e.g., where the dynamics or the rewards vary over different trials of experiments and cannot be controlled by humans. We will clarify when the two-point method can be used in practice, when it cannot be used and only one-point method can be used in our final revisions.
> > > > >
> > > > > 4. Thanks for your suggestion. We will use the term bias in gradient estimator to describe the difference between the estimator in [Bach & Perchet, 2016] and ours.
> > > > >
> > > > > 5. Thanks for your suggestion. In the final revisions, we will clarify the meanings of all the expectation signs in our paper.
> > > > >
> > > > > We want to thank the reviewer for your patience and all the valuable advice. All of your suggestions will definitely be reflected in our final revisions of the paper.

---

> > > ### Author Response · Authors · 2020-11-23
> > > **Response to Reviewer 1's feedback**
> > >
> > > We thank the reviewer for your response.
> > >
> > > We first clarify a big concern of the reviewer regarding the random sequence of functions {$f_t$}. We emphasize that their generation is assumed to be independent from the variable $x$ and perturbation $u$. We have formally stated this after introducing the problem (P). Similarly, the random function evaluation noises $\xi_t$ are also independent from the variable $x$ and perturbation $u$. We have formally stated this after problem (R). These independence assumptions are satisfied, e.g., by the function we give under (8), or the numerical examples in Section 6.
> > >
> > > Below is our point-to-point response:
> > >
> > > 1. In the proof of Lemma 2.5, the reviewer seems to be concerned about the last term in (14). We would like to emphasize that in this term, the argument in $f_t$ does not depend on $u_t$. Instead, since the functions {$f_t$} are randomly generated independently of the agent’s decisions, the random variable $f_t(z_{t-1}) - f_{t-1}(z_{t-1})$, where $z_{t-1} = x_{t-1} + \delta u_{t-1}$, is independent of the random search directions $u_t$. Therefore, according to the fact that $E[XY] = E[X]E[Y]$ when $X$ and $Y$ are independent, the bound provided in the lines below (14) holds. This has been clarified under (14). Similarly, the proof of Lemma 5.2 is also clarified under (42).
> > >
> > > 2.a. We have now clarified the problem statements under (P) and (R), and the related proofs.
> > >
> > > 2. b. The reviewer is correct that the expectation in (33) is taken conditional on $x_t$. Then, we use the tower rule of conditional expectation to obtain (34), where the expectation is a full expectation. We have clarified how we achieve the transition in the lines between (33) and (34).
> > >
> > > To obtain the results in Theorem 4.2 and 4.3, we did not use Assumption 3.1. However, in the previous proofs of these two theorems, we showed more results than those that we provided in the main content of the paper. Specifically, we presented two versions of our results under different combinations of assumptions, i.e., when both Assumption 4.1.1 and 3.1 hold, and when Assumption 4.1.2 holds but not Assumption 3.1. We apologize for this confusion. Now we have removed the results we obtained using Assumption 3.1 in the proofs of these two theorems.
> > >
> > > 2. c. The proof of Theorem 4.2 used Assumption 4.1.2 to obtain the inequality (35). This essentially sums the term $D_t$ in (5) over time so that the bound in Assumption 4.1.2 can be applied. If the reviewer agrees that Lemma 2.5 is correct after our clarification, then the bound (45) is also correct.
> > >
> > >
> > > 3. We apologize for the confusion in the notation. $V_{f,\xi}^2$ is not a random variable related to $\xi_t, \xi_{t-1}$. Instead, it is a finite number that can bound the variation of the function together with its evaluation variance.
> > >
> > > 4.  Thanks for the comments. The condition that “we can still run two trajectories in parallel by applying two controls” on the same fixed dynamical matrices $(A_t, B_t)$ is only possible in a simulated environment, where the evolution of $(A_t, B_t)$ to the next pair can be decided by the user. In practice, this condition is difficult to satisfy, because the agent is not able to control when the system evolves to the next pair of dynamical matrices. When a policy $K_t + \delta U_t$ is implemented on a system with (A_t, B_t), the system evolves for $H$ time steps under control policy $K_t + \delta U_t$ and subject to a single sequence of noise $\{w_k\}$. Then, the evaluation of $V_t$ is obtained at the end of episode $t$, and the system switches to the new dynamics, no matter whether we want it to change or not. During episode $t+1$, the agent can evaluate a policy only for the new function $V_{t+1}$ in practice, not the past function $V_t$.
> > >
> > > To summarize, the system varies over episodes, each episode takes $H$ time steps, and the system will change at the end of each episode no matter whether we want it to change or not. Evaluating the same function $V_t$ at two different controls suggest that we can move back in time and do counterfactual evaluation: what if I did $K_t - \delta U_t$ instead of $K_t + \delta U_t$ during episode $t$. This is only possible in simulated environments where the user has full control of the system. In practice, this can never happen.

---

### Official Review · AnonReviewer2 · 2020-10-28
**Replacing one of the two function samples in zeroth-order online learning algorithms with an old sample collected at a previous iteration: potential and limitations of the technique.**

**Rating:** 4
**Confidence:** 4

**Review:**

This paper proposes a zeroth-order (derivative-free) algorithm for online stochastic optimization problems. The objective is to find a sequence of actions $x_0,\dots,x_{T-1}$ minimizing the expected regret
$$\mathbb{E} [ \sum_{t=0}^{T-1} f_t(x_t) - \min_{x\in\mathcal{X}} \sum_{t=0}^{T-1} f_t(x)],$$ where the (sub-)gradients of unknown cost functions $f_t$ are not available, and only measurements $f_t(x_t)$ of the values of the functions at tests points $x_t$ can be obtained.

The submission builds on the zeroth-order techniques developed by Nesterov & Spokoiny in [1] for derivative-free, non-smooth, convex and non-convex optimization, where similar gradient estimation techniques based on sampling and Gaussian smoothing are used, with the difference that two values of an identical noisy instance of the cost function are needed in [1] at each iteration. By requiring only one noisy function value per iteration and recalling the function value collected during the previous iteration (in the submission this technique is called "residual feedback"), the proposed algorithm extends convergence results of the two-point approach [1] to regret bounds in stochastic/bandit settings where the function is changing after every new value observed, on condition that the differences between two consecutive instances of the cost function are bounded in variance.

The regret bounds derived in the paper match those obtained for recent 'one-point' zeroth-order methods for online optimization (e.g. [2]). A specificity of the algorithm proposed in the paper, compared to other 'one-point' methods, is that the algorithm does not depend on the absolute function levels, only on differences between two function instances, which may improve the performance in practice, as shown in the numerical experiments. This property was also shared by the approach of Bach and Perchet [4], which serves as a benchmark algorithm in the numerical experiments of the submission. These experiments are carried out on a nonstationary LQR control algorithm, and on a nonstationary resource allocation problem.

The paper is technically sound and the developments are clear. The regret bounds derived for online non-convex optimization are interesting. The contributions to the online convex optimization framework are less obvious, due to the abundant literature on the topic. See my concerns below and my questions to the authors.

I look forward to the authors' answers. My recommendation will be amended after their rebuttal.



Pros:

The paper is technically sound and well written.

The regret bounds derived in the non-convex online optimization framework are of particular interest.

Since the proposed algorithm does not depend on the function levels, it may perform better than the basic 'one-point' methods in practice.



Concerns and questions:

The presentation of the results leaves a mixed impression. I agree that regret bounds in non-convex online learning are a contribution to the field. The claims of novelty made by the authors for the convex case, on the other hand, look somewhat overstated. They write, for instance: "it is also the first time that a one-point gradient estimator demonstrates comparable performance to that of the two-point method". This sounds optimistic to me, in the sense that the authors' argument is mostly empirical (numerical experiments for a particular problem), whereas the regret bounds derived in the paper do not compare with the regret bounds that two-point methods would achieve. Moreover, there exist more recent approaches to convex zeroth-order online learning which claim the conjectured $\Omega(\sqrt{T})$ regret bound [3,5]. These new trends in zeroth-order online learning are not discussed in the submission.

I don't see a clear distinction between the settings 'online bandit optimization' (Sections 3 and 4) and 'online stochastic optimization' (Section 5), because the regret criterion (6), the assumptions of the cost function sequences (3.1, 4.1 / 5.1, 5.2), the algorithms, and the regret bounds are apparently the same for the two settings. The only specificity of the Section 5 model seems to be the existence of a mean cost function $\mathbb{E}[f_t]$, if we set $f_t(\cdot)\equiv F(\cdot;\xi_t)$ — assumption which is not exploited. Also, I found Section 5 slightly redundant. I thought it could easily be replaced by a discussion on all the frameworks covered by Assumptions 3.1 and 4.1 and on the possible interpretations given to the model.

In the submission, an algorithm developed by Bach and Perchet in [4], was classified by the authors as a two-point zeroth-order optimization algorithm and used in the numerical experiments as a benchmark for comparison. In my recollection of [4], the algorithm relies on a gradient estimator which considers the difference between two noisy functions values affected by two independent noises, with the assumption that the noises are uniformly bounded in variance or satisfy a martingale property. To me, these assumptions are similar (if not identical) to those made in Section 5 and in Sections 3,4,6, respectively. Could the authors clarify the differences between the noise model of [4] and the one they use, and why the algorithm [4] is impractical and cannot be used in online settings? Why was it treated differently in the numerical experiments?

Another feature that was not discussed in the paper is the feasibility of the algorithm in terms of the availability of the function queries. The problem stated in Equation (P) is a constrained online optimization problem over a convex set. However, since the test points are sampled over the entire state space from Gaussian distributions, the proposed algorithm will query function values outside the feasible set, and these function values are not available in many learning applications. Note that it is possible to combine Gaussian sampling with constrained online optimization [5], and that feasible zeroth-order optimization algorithms based on residual feedback have been developed [6].



Typos :
p.2 such an one-point derivative-free setting => such a
p.7 nonstatinoary => nonstationary

[1] Yurii Nesterov and Vladimir Spokoiny. Random gradient-free minimization of convex functions. Foundations of Computational Mathematics, 17(2):527–566, 2017.

[2] Alexander V Gasnikov, Ekaterina A Krymova, Anastasia A Lagunovskaya, Ilnura N Usmanova, and Fedor A Fedorenko. Stochastic online optimization. single-point and multipoint non-linear multi-armed bandits. convex and strongly-convex case. Automation and remote control, 78(2):224–234, 2017.

[3] https://arxiv.org/abs/1603.04350

[4] Francis Bach and Vianney Perchet. Highly-smooth zero-th order online optimization. In Conference on Learning Theory, pp. 257–283, 2016.

[5] https://arxiv.org/abs/1607.03084

[6] https://arxiv.org/abs/2006.05445


__________



Update after the discussions:

I would like to thank the author(s) for all their comments. Although most of my concerns have been addressed, some questions remain topics of contention. Before discussing these topics, I will first append to this review my answer to the author(s)' last comments, as it was their wish to keep hearing from me after closing of the discussions:

$\ \ \ $ The assumptions (3.1, 4.1, 5.1, 5.3) made on the function sequence $f_t$ for convergence of the proposed algorithm are unconventional as they require that the expected absolute variations of two function values at the points visited by the algorithm be bounded, or that the squared variations of two function values obtained by Gaussian sampling from points visited by the algorithm be bounded. So formulated, the conditions for convergence involve the algorithm's trajectory $x_t$ as much as the function sequence $f_t$, and they are difficult to verify. In an attempt to identify sufficient conditions for these assumptions to hold true, I made three suggestions: (i) and (ii) were concerned with the boundedness of the sequence of points generated by the algorithm, and (iii) was the case of bounded incremental variations of the sequence $f_t$, e.g. martingales. In their reply, the author(s) were right to rule out (i) and (ii), which indeed were unrelated. This leaves us with (iii) as a possible setting for the proposed algorithm.

$\ \ \ $ In my last comment I argued that the case (iii), where the sequence $f_t$ undergoes incremental variations uniformly bounded in expectation, was covered by the approach taken in Bach & Perchet (2016), where two function queries obtained from perturbations around the same iterate are processed at each step. The Bach/Perchet approach is cited in the paper for comparison, but it is called impractical as it would not apply when $f_t$ varies over time $-$ argument I disagree with and that I attempted to refute in a brief discussion involving martingale-like variations for $f_t$. When the author(s) of the submission object to my regret analysis in the case of martingale-like noise on the basis that the assumptions they make also cover non-zero-mean variations with similar uniform upper bounds on the moments, they do not address the main point of my comment. My intention was to show that it does not take much effort to consider the approach used in Bach & Perchet (2016) in settings where the cost function is changing over time, for as long as the cost variations are incremental with bounded moments. This can be seen by noting that the convergence result derived in the revised version of the supplementary material for the residual-feedback algorithm with unit-sphere sampling can be reproduced for the Bach/Perchet approach under the considered assumptions. I take it that the author(s), who excel at deriving the convergence rates for such algorithms, will not disagree. Although the assumptions used in Bach & Perchet (2016) (uniform zero-mean increments) may look somewhat stricter, they have the merit of being clear and simple, as opposed to Assumptions 3.1, 4.1, 5.1, 5.3, which involve the trajectory of the algorithm and can't be verified easily. They are also sufficient to improve the convergence rates for higher degrees of smoothness compared to the early algorithm by Flaxman et al. (2005), which was the objective of that paper. Higher degrees of smoothness failing which it is difficult to improve the convergence rates, as confirmed by the convergence rates given in the submission. In my sense, one important message conveyed by the submission is that the approaches proposed in the submission and in Bach & Perchet (2016) can both handle bounded additive noise, and both fail in the more general framework of adversarial learning. By calling the Bach/Perchet algorithm impractical for their setting, I believe the author(s) of the submission missed to chance to compare the two approaches from a fair perspective and to answer the simple question that comes to mind when reading their paper: is the residual feedback technique really useful in the stochastic learning framework, or isn't convergence just as fast when the function queries are processed by pairs as in Bach & Perchet (2016), or in the reference paper by Nesterov & Spokoiny (2017) ?

-------

That being said, the following issues remain in this submission:

$\bullet$ The assumptions (3.1, 4.1, 5.1, 5.3) made on the function sequence $f_t$ for convergence of the proposed algorithm are unconventional and difficult to verify, because they consist in properties of the iterates of the algorithm.

$\bullet$ In our discussions, only incremental sequences $f_t$ with variations uniformly bounded in expectation have been identified to meet those assumptions. In my sense, this particular setting is also covered by the approach taken in Bach & Perchet (2016), where the function queries are handled by pairs obtained from perturbations around the same iterate. Also, I still find it unfair to call the latter approach impractical for the considered setting.

$\bullet$ Since the convergence rates derived in the paper show no clear improvement, compared to the early approach of Flaxman et al. (2005), the arguments of the submission lie in the experimental results, where I don't think the algorithm by Bach & Perchet (2016) is given a fair treatment (for the reason explained in the previous paragraph). Besides, the application considered in Section 6.1 reduces to the unconstrained minimization of a polynomial of high degree that is neither Lipschitz nor smooth, which is a basic requirement for the convergence algorithms. This makes the convergence of the algorithms highly dependent on the initial point, unless optimization is done over a compact set, but I don't think the projection step was implemented for the algorithms.

$\bullet$ In constrained optimization, the problem that the proposed algorithm samples function values outside the feasible set has been partly addressed by the author(s), who provided a variant of the algorithm based no longer on Gaussian sampling, but on sampling over a sphere. Partly because only one convergence result for a particular setting was derived, and it remains unclear (as pointed out by Reviewer 4) if all the benefits of Gaussian smoothing and all convergence results would also extend to spheric smoothing. This discussion is missing. In my opinion, the extension to settings where the functions can't be sampled outside the feasible set is not absolutely imperative in all frameworks (the author(s) have provided counter-examples), but it would be useful to know the limits of the proposed technique.

All things considered, I would not recommend the submission for presentation at the conference. Independently of the final decision, I hope the author(s) will make the most from the discussions with all the reviewers.

I would like to make a last comment about the submission and the discussions that followed. It is natural that the author(s) give the best picture of the algorithm they propose. Yet in the paper the contrast is particularly strong between, on the one hand, the haziness surrounding the assumptions made on the function sequence $f_t$, or the negligence with which the algorithms were applied in Section 6.1 to a problem not actually meeting the conditions for convergence, and on the other hand the severity with which the Bach/Perchet approach was disqualified as a possible method of solution. This contrast gives the reader an overall feeling of partiality, which makes the reviewing task an intricate, contradictory, and unappreciative one.

---

> ### Author Response · Authors · 2020-11-21
> **(Cont'd) Response to Reviewer 2**
>
> Q6: Discussion on the ability to query function values outside the constraint set.
>
> A: We thank the reviewer for raising this question. In this paper, we follow the existing literature and assume that $f_t$ can be evaluated outside the constraint set. This assumption is widely adopted, e.g., in the following literature [A-E]. In particular, we note that the constraint set only restricts the final optimization solution, and it does not restrict the feasibility of the function evaluations.
>
> On the other hand, we also want to mention that with minor modifications we can guarantee the feasibility of the perturbed points. We elaborate on these modifications in the Section H in the supplementary material. In short, we can use unit sphere sampling instead of Gaussian sampling, and perform the projection onto a shrinked constraint set $(1-\xi)\mathcal{X}$ for some small $\xi>0$ (see eq.(48)), where the shrinkage coefficient $(1-\xi)$ is to make sure that the perturbed point is within the original set $\mathcal{X}$.
>
> [A] Duchi, et al, A. (2015). Optimal rates for zero-order convex optimization: The power of two function evaluations.
>
> [B] Bach, F., & Perchet, V. (2016). Highly-smooth zero-th order online optimization.
>
> [C] Liu, et al, (2018). Zeroth-order stochastic projected gradient descent for nonconvex optimization.
>
> [D] Balasubramanian, K., & Ghadimi, S. (2018). Zeroth-order nonconvex stochastic optimization: Handling constraints, high-dimensionality and saddle-points.
>
> [E] Sahu, A. K., & Kar, S. (2020). Decentralized Zeroth-Order Constrained Stochastic Optimization Algorithms: Frank-Wolfe and Variants With Applications to Black-Box Adversarial Attacks.
>
> We also thank the reviewer for pointing out the solutions in [5,6]. The discussion of [5] is included in our response to Q3. In [6], the authors studied a similar oracle for a static convex optimization problem with objective and constraint functions of specific forms. The above discussion and the related works mentioned by the reviewer have been included in the revision under update (4).

---

> > ### Comment · AnonReviewer2 · 2020-11-24
> > **Additional question about your Assumptions, which I fail to understand.**
> >
> > Many thanks. All your replies are much appreciated.
> >
> > I would like to ask the author(s) for further clarification on the assumptions they make with regard to the sequence of functions $f_t$, hoping there is still time for them to answer. I now realize that I had misread those assumptions when I asked Rev2.Q4 and Rev2.Q5.
> >
> > In answer to Rev4.Q2, the author(s) have made clear, in the revised version of the the submission, that the expectations considered in Assumptions 3.1, 4.1, 5.1, 5.3 are taken with respect to the joint distribution of the variables $u_{t-1}$,$f_{t-1}$, $f_t$, $x_{t-1}$, $x_t$, where the sequence $(x_t)$ is the sequence of points generated by the algorithm. I am struggling to see how Assumptions 3.1, 4.1, 5.1, 5.3 can be verified, unless maybe one of the following holds:
> >
> >   $(i)$ the feasible set $\mathcal{X}$ is compact, in which case the sequence $(x_t)$ is bounded,
> >
> >   $(ii)$ the sequence $(x_t)$ generated by the algorithm can be shown without these assumptions to converge with high probability (in which case the sequence $(x_t)$ is also bounded),
> >
> >   $(iii)$ the variations of the functions are bounded (in expectation) uniformly for $x \in \mathbb{R}^d$.
> >
> > Condition $(i)$ is more restrictive than the initial assumption made in the submission that the feasible set is merely convex. Condition $(ii)$ may not hold either, since all the convergence results derived in the study for the sequence $(x_t)$ seem to rely on Assumptions 3.1, 4.1, 5.1, or 5.3.  And $(iii)$ is the case when the expectations hold conditionally on $(x_l)_{l=1}^{t}$, so that the sequence $(f_t)$ is martingale-like, with uniformly bounded increments, but this is not what is meant by the author(s) in their revised submission.
> >
> > Does anyone (author or reviewer) have an explanation on how Assumptions 3.1, 4.1, 5.1, or 5.3 can be verified in practice?

---

> > > ### Author Response · Authors · 2020-11-24
> > > **Response to Reviewer 2**
> > >
> > > We thank the reviewer for raising this question that helps us better clarify these  assumptions.
> > >
> > > If we understand the reviewer’s comments correctly, the reviewer proposed several conditions that can potentially imply our assumptions. Regarding the conditions (i) and (ii), we note that bounded sequence {$x_t$} does not imply our Assumption 3.1, as the function variation $f_t(x)-f_{t-1}(x)$ can still be arbitrary large for a bounded $x$. Regarding the condition (iii), it says that there exists $V_f^2>0$ such that $E\big[ \| f_t(z) - f_{t-1}(z)\|^2 | z\big] \leq V_f^2$ for all $z \in \mathbb{R}^d$. We agree that this condition can imply our Assumption 3.1. Furthermore, if $E\big[ f_t(z) - f_{t-1}(z) | z \big]$ is bounded by a constant for all $z$, then the total expectation $E \big[ f_t(z) - f_{t-1}(z) \big]$ is also bounded. These conditions suffice to verify our Assumption 4.1, which bounds the summation of the bounded variation of functions in expectation over a finite period of time.
> > >
> > > If we understand reviewer’s condition (iii) correctly, in the next, we present some practical examples where the conditional expectation mentioned above is bounded, which further suggests that our Assumptions can be verified in these examples.
> > >
> > > Example 1: Time-varying objective function with additive noise. This example is presented after (8). Consider the initial function $f_0(x) = 1/2x^2$, and let $f_{t} = f_{t-1} + n_t$, where $n_t \sim \mathcal{N}(0, \sigma^2)$ is independent from all other variables. Then, it is easy to check that the conditional expectation is bounded for all x. And Assumption 3.1 holds with $V_f^2 = \sigma^2$.
> > >
> > > Example 2: Episodic RL with varying reward functions. Consider an episodic RL problem with horizon $H$ and a fixed transition function over episodes. Then, the value function of a policy $\pi$ at episode $t$ is defined $V_t(\pi) = \sum_{k = 1}^H r^t(s_k, a_k)$, where {$(s_k, a_k$} are state-action trajectories generated by implementing policy $\pi$. Since the transition function is fixed, the state-action trajectories are the same when evaluating policy $\pi$ at two episodes. If the reward function varies as $r^t(s_k, a_k) = r^{t-1}(s_k,a_k) + n^t_k$, where $n^t_k$ is a random variable having a bounded support or is bounded in its first and second order moments, e.g., Gaussian noises. Then, $V_{t+1}(\pi) - V_t(\pi) = \sum_{k=1}^H n^t_k$ which could also be subject to bounded first and second order moments. Then, it is easy to check that the conditional expectation is bounded for all pi. Therefore, Assumptions 3.1 holds.
> > >
> > > Assumption 4.1 essentially measures the accumulated variation between different time steps. If the variation between two consecutive time steps is bounded, then the accumulated variation over a finite period of time is also bounded. Specifically, let Assumption 3.1 hold with a finite time-varying bound $V^2_{f,t}$. Then, the constant $\widetilde{W_T}$ in Assumption 4.1.2 is the summation of $V^2_{f,t}$. Therefore, the examples listed above also showcase when Assumption 4.1.2 holds. Similarly, the above examples also suggest that Assumption 4.1.1 holds.
> > >
> > > Assumptions 5.1 and 5.3 consider the additional evaluation noises $\xi_t$. A simple example to satisfy these two assumptions is when the evaluation noises are additive. Consider Example 3 above, if the reward function is defined as $r^t(s_k, a_k) = r^{t-1}(s_k,a_k) + n^t_k + w_k$, where $w_k$ is a Gaussian noise independent of the state, the action and the policy. Then, $\sum_{k=1}^H w_k$ constitutes the evaluation noise $\xi_t$. It is straightforward to see that Example 3 with this additional evaluation noises satisfy Assumptions 5.1 and 5.3.
> > >
> > > The above example list is of course not an exhaustive list. We hope these examples can help clarify when these assumptions are satisfied. We would also like to mention that the bounded conditional expectation is only a sufficient condition to verify our assumptions. Our assumptions can be potentially satisfied in more general scenarios.

---

> > > > ### Comment · AnonReviewer2 · 2020-11-25
> > > > **About your examples.**
> > > >
> > > > ​Thank you for your quick reply.
> > > >
> > > > About your examples:
> > > >
> > > >
> > > > $\bullet$
> > > > In my understanding of your Example 2, if we follow the instructions of Section 6.1 with $u_k = K x_k$, we get
> > > > $$
> > > > x_{k} = A_t x_{k-1} + B_t u_{k-1} + w_{k-1} = ( A_t + B_t K) x_{k-1} + w_{k-1} =  ( A_t + B_t K)^{k} x_0 +  \sum_{l=0}^{k-1} ( A_t + B_t K)^{k-l} w_l
> > > > $$
> > > > and the cost function "$f_t$" is given by
> > > > $$
> > > > V_t (K)
> > > > :=
> > > > \mathbb{E} [ \sum_{k=0}^{H-1} \gamma^k (x_k^T Q x_k + u_k^T R u_k ) ]
> > > > =
> > > > \mathbb{E} [ \sum_{k=0}^{H-1} \gamma^k (x_k^T (Q + K^T R K) x_k ) ]
> > > > =
> > > > P_{2H}(K)
> > > > $$
> > > > which is a polynomial of degree $2H$ in the unknown matrix $K$, and of degree $2H-2$ in the random parameters $A_t$ and $B_t$.
> > > >
> > > > If the feasible set $\mathcal{X}$ of the variable $K$ is the whole matrix space (the problem is unsconstrained), then the  cost function is not Lipschitz continuous, which is a basic assumption in your paper.
> > > >
> > > > If otherwise the $\mathcal{X}$ is a compact set, then indeed bounds on expected differences of the type $\mathbb{E}[|f_t(x_{t-1}+\delta u_{t-1})-f_{t-1}(x_{t-1}+\delta u_{t-1})|^2]$ follow from bounded variations of the random parameters $A_t,B_t$. In that case, it is unclear to me if really the so-obtained upper bound is less restrictive than the uniform bound $\mathbb{E}[|f_t(\cdot;\xi_{t-1})|^2]\leq B$ used by Gasnikov et al. (2017).
> > > >
> > > >
> > > >
> > > > $\bullet$ I believe your Example 1 is additive bounded noise, while the remaining examples consider uniformly bounded increments of the type
> > > > $$ f_t=f_{t-1}+\eta_{t}=f_0+S_t,$$
> > > > where $S_t=\sum_{l=1}^{t}\eta_l$ defines a martingale. Both cases are covered by the setting of Bach & Perchet (2016) [cf. discussion at the bottom of p. 2 in Bach & Perchet (2016)]. Now, in the noise model described by the above equation, we have, for the expected regret,
> > > > $$
> > > > R_T := \mathbb{E} [ \sum_{t=0}^{T-1} f_t(x_t) - \min_{x\in\mathcal{X}} \sum_{t=0}^{T-1} f_t(x)]
> > > > = \mathbb{E} [ \sum_{t=0}^{T-1} (f_0(x_t)+S_t) - \min_{x\in\mathcal{X}} \sum_{t=0}^{T-1} (f_0(x)+S_t)]
> > > > = \mathbb{E} [ \sum_{t=0}^{T-1} f_0(x_t) - \min_{x\in\mathcal{X}} \sum_{t=0}^{T-1} f_0(x)]
> > > > := R_{0,T}
> > > > ,
> > > > $$
> > > > and we can define
> > > > $$
> > > > \tilde g_{t-1}(x_{t-1}) :=
> > > > \frac{u_{t-1}}{\delta} [f_{t} (x_{t-1} + \delta u_{t-1} ) - f_{t-1} (x_{t-1}  - \delta u_{t-1} )]
> > > > =
> > > > \frac{u_{t-1}}{\delta} [f_{0} (x_{t-1} + \delta u_{t-1} ) - f_0 (x_{t-1}  - \delta u_{t-1} ) + \eta_{t}],
> > > > $$
> > > > which is exactly (5) in Bach & Perchet (2016).
> > > > Hence, the algorithm of Bach & Perchet (2016) applies in this case by collecting two (different) functions $f_t$ and $f_{t+1/2}$ at each time step, and using the gradient estimator $$
> > > > \hat g_t(x_{t}) :=
> > > > \frac{u_{t}}{\delta} [f_{t+1/2} (x_{t} + \delta u_{t} ) - f_{t} (x_{t}  - \delta u_{t} )]
> > > > .$$ By proceeding so, we obtain a bound on the regret $R_{0,2T} = R_{2T} $.
> > > > On the one hand, the regret bound is twice greedier in terms of function queries. On the other hand, the  algorithm may converge faster because the gradient estimator is more accurate than with the residual feedback, since the two function values are now computed at $x_{t} \pm \delta u_{t}$, no longer at $ x_{t} + \delta u_{t}$ and $x_{t-1} + \delta u_{t-1}$ as in the residual-feedback gradient estimator. All in all, the algorithm Bach & Perchet (2016) can be implemented, and it is difficult to say from this discussion which algorithms performs better.
> > > >
> > > > In summary: it is still unclear to me when the assumptions made in the submission on the sequence $f_t$ apply, how they can be verified in the general case, in what sense they improve on the other assumptions made in the literature (e.g., Gasnikov et al. (2017) or Bach & Perchet (2016)), or when the algorithm by Bach & Perchet (2016) is impractical in online settings.

---

> > > > > ### Author Response · Authors · 2020-11-25
> > > > > **(Cont'd) Response to Reviewer 2's feedback**
> > > > >
> > > > > 4. On the comment that $f_t = f_0 + S_t$, where $S_t$ is a martingale.
> > > > >
> > > > > We want to point out that the examples we gave (Example 1 and 2 in our updated reply in the last round) do not necessarily suggest $S_t$ is a martingale. Note that $S_t$ is a martingale only when $\eta_t$ is zero mean for all $t$. However, in our examples, we do not require the random variation term $\eta_t$ to be zero mean. To see this, in Example 1, the variation term $n_t$ can be $n_t \sim \mathcal{N}(1, \sigma^2)$. This still satisfies our Assumptions 3.1 and 4.1. Similarly, in Example 2 (previously it was Example 3), $\sum_{k=1}^H n_k^t$ can also be non-zero mean as long as it is of bounded first and second order moments.
> > > > >
> > > > > We also want to clarify that the time-varying objective function is not simply moving upwards or downwards in parallel. This seems to be the understanding of the reviewer from his/her derivation in $R_T$. To see this, in Example 2, if $V_t$ is evaluated at two different policies $\pi$ and $\pi’$, since $\pi$ and $\pi’$ lead to different state-action trajectories, the magnitude of the variation $\sum_{k=1}^H n_k^t$ can also be different at $\pi$ and $\pi’$. Recall that $n_k^t$ corresponds to the change of the reward from $r^{t-1}$ to $r^t$ at $(s_k, a_k)$. This means that at different state-action trajectories generated by different policies, the variation of the total rewards from $t-1$ to $t$ could be different in expectation. Therefore, the derivation of $R_T$ is not generally correct.
> > > > >
> > > > > Following the above clarification, there are two mistakes that we noticed in the derivation of $\tilde{g_{t-1}}$. First, the change from $f_0$ to $f_t$ (or $f_{t-1}$)  at the points $x_t + \delta u_t$ (or $x_t - \delta u_t$) should in fact be denoted as $S_t(x_t + \delta u_t)$ (or $S_{t-1}(x_t - \delta u_t)$), because our assumptions do not assume the variation of the function value at different points over time is the same. Then, $S_t(x_t + \delta u_t) - S_{t-1}(x_t - \delta u_t) \neq \eta_t$. Even in the case that variation of the function value at different points over time is the same, and we can obtain $S_t(x_t + \delta u_t) - S_{t-1}(x_t - \delta u_t) = \eta_t$, since $\eta_t$ is not zero-mean, the analysis in [Bach & Perchet, 2016] cannot be used. Because they require $\eta_t$ to be zero-mean. To see this, the reviewer can refer to the definition of $\epsilon_n$ in the bottom of p2 in [Bach & Perchet, 2016], or the bottom of p7 in [Bach & Perchet, 2016].
> > > > >
> > > > > Due to the above reasons, the regret bound of the oracle $\hat{g}_t(x_t)$ proposed by the reviewer cannot be directly analyzed using the regret analysis in [Bach & Perchet, 2016], because they require their bias term $\epsilon_n$ (or $\eta_t$) to be zero mean.

---

> > > > > ### Author Response · Authors · 2020-11-25
> > > > > **Response to Reviewer 2's feedback**
> > > > >
> > > > > We greatly thank the reviewer for the quick and informative response. Below are our quick comments. Although this might be our last response, we are still willing to hear from the reviewer.
> > > > >
> > > > > 1. In what cases the assumptions are satisfied
> > > > >
> > > > > We have updated our last reply to describe a sufficient condition that can satisfy our assumptions and also present some examples when this sufficient condition is satisfied. We hope this now becomes more clear.
> > > > >
> > > > >
> > > > > 2. Regarding the old example 2 we provided.
> > > > >
> > > > > First, we would like to clarify that the discussion below only limits to the question whether a very specific learning application can satisfy the assumptions in this paper. It does not affect the correctness of any results built on these assumptions for other applications.
> > > > >
> > > > > Second, we have deleted the old example in our last reply before we saw the reviewer’s last response, because we realized that it does not satisfy the bounded conditional expectation condition mentioned by the reviewer, when $K$ can be selected over the whole domain. But we are still willing to discuss with the reviewer about some of our thoughts on how to apply our results in this specific setting.
> > > > >
> > > > > As the reviewer mentioned, one issue is that when $K$ can be selected over the whole domain, the function $f_t$ is not globally Lipschitz. In this case, if there exists a uniform Lipschitz constant that can be used over the random iterates of the algorithm (with a high probability), e.g., see the transition from (15) to (16), the Lipschitz constant only needs to explain the change of the value from $x_{t} + \delta u_t$ to $x_{t-1} + \delta u_t$, then the bounds still hold. However, such proof has not been rigorously constructed for this specific application. And we agree with one of the reviewer’s previous intuitions, that in this scenario it may be possible to show convergence with high probability. Although we have not built a theoretical proof for this very specific example, our numerical experiment shows that our proposed oracle can significantly outperform the conventional one-point method.
> > > > >
> > > > > 3. On the comparison with Gasnikov’s assumption
> > > > >
> > > > > As we mentioned before, we require that the variation between $f_t$ and $f_{t-1}$ is bounded in expectation, while Gasnikov assumes $f_t$ is uniformly bounded over time. Since the sequence of {$f_t$} satisfies Gasnikov’s assumption must satisfy ours, while the sequence that satisfies ours does not necessarily satisfy Gasnikov’s assumption, our assumption is weaker than Gasnikov’s assumption.
> > > > >
> > > > > And we would also like to clarify that our proposed method improves from conventional one-point method considered by Gasnikov’s works not just from weaker assumption perspective, but also from the fact that it has much better empirical performance than conventional one-point method’s performance in this non-stationary setting. We believe both the weaker theoretical assumption in the analysis and its much better practical performance should be recognized.

---

> ### Author Response · Authors · 2020-11-21
> **Response to Reviewer 2**
>
> We thank the reviewer for providing valuable feedback that helps us improve the quality of this paper. Below is a point-to-point response to the reviewer’s concerns.
>
> Q1: The claims of novelty made by the authors for the convex case look overstated.
>
> A: We agree that the statement is not precise. In the contribution section, we have changed the statement to “it is also the first time that a one-point gradient estimator demonstrates comparable **empirical** performance to that of the two-point method”.
>
> Q2: The regret bounds derived in the paper do not compare with the regret bounds that two-point methods would achieve.
>
> A: In this study, our focus is online optimization in dynamic environments, where the conventional two-point method is not applicable. The reason for this is discussed in our response to Q5 below. Therefore, our developed one-point method cannot be directly compared to the conventional two-point method.
>
> Q3: There are more recent approaches [3,5] to convex zeroth-order online learning that are not discussed in the submission.
>
> A: We thank the reviewer for pointing out these related works. We have cited and discussed them in the revision.
>
> To elaborate, these two works study online bandit algorithms using ellipsoid methods. In particular, these methods induce heavy computation per step and achieve regret bounds that have poor dependence on the problem dimension. As a comparison, our one-point method is computationally light and achieves regret bounds that have better dependence on the problem dimension.
>
> Q4: No clear distinction between the online bandit optimization (Sections 3 and 4) and online stochastic optimization (Section 5). Section 5 is slightly redundant and can be replaced by a discussion.
>
> A: We note that Sections 3 & 4 consider noiseless online function evaluations,  whereas Section 5 considers the more general noisy online function evaluations. Therefore, the residual feedback estimator and the assumptions in Section 5 are adapted to the noisy setting accordingly.
>
> In fact, our presentation of Section 5 is indeed a general discussion. In the beginning of Section 5, we mention that “Since its regret analysis follows the same proof logic as that of ZO with residual feedback, we only introduce the key technical lemmas and comment on the proof difference.” Our Section 5 only outlines the problem setup, estimator design and assumptions.
>
> Q5: Clarify the difference between the noise model of [Bach & Perchet, 2016] and the one they use. Why is it impractical in online settings and treated differently in the numerical experiments?
>
> A: We thank the reviewer for raising this question. We first clarify the difference between their estimator and ours. The gradient estimator (7) in [Bach & Perchet, 2016] queries the same function $f_t$ twice at two different points. As a comparison, our residual feedback queries $f_t$ at a single point and utilizes the feedback $f_{t-1}$ obtained from the previous time step. Hence, the estimator in [Bach & Perchet, 2016] cannot be applied to online problems in dynamic environments, where each function $f_t$ can **only be queried once**.
>
> To further elaborate on the impracticability of the estimator in [Bach & Perchet, 2016], consider the non-stationary online LQR experiment in Section 6, where the dynamic matrices $A_t, B_t$ are random and time-varying over the episodes $t$. Evaluating the value function $V_t$ at a given policy requires to collect samples during episode $t$ by implementing this policy. Then, in the subsequent episode $t+1$, due to the non-stationarity of the problem, the dynamic matrices evolve to $A_{t+1}, B_{t+1}$ and define a new value function $V_{t+1}$. Hence, the dynamic environment does not allow to evaluate the same value function $V_t$ at two different episodes and, therefore, two-point methods are not applicable here. We have highlighted this discussion in Section 6.1.
>
> We further note that the noise model adopted in [Bach & Perchet, 2016] is different from ours in our Assumptions 3.1, 4.1, 5.1 and 5.3. The function evaluation noise $\epsilon_n$ at time $n$ in [Bach & Perchet, 2016] is assumed to be zero-mean. As a comparison, our Assumptions 3.1, 4.1, 5.1 and 5.3 do not require the noise to be zero-mean, and instead we bound the time variation of the objective function.

---

### Official Review · AnonReviewer4 · 2020-11-01
**Recommendation to reject**

**Rating:** 4
**Confidence:** 4

**Review:**

This manuscript considers online zeroth order optimization and it develops a gradient estimator based on one query per function. In particular, the proposed method mimics two-point estimators by evaluating two consecutive functions at perturbations of an iterate, as shown in equation (3). Although one-point gradient estimates are possible, they have impractically large variances. Given this limitation and the wide need of zeroth order optimization (in particular in RL), the study of two-point estimators is important.

While this manuscript has many strengths, there are several issues that need to be clarified before it can be accepted for publication.

Pros:
- This work offers a simple solution.

- Also, the authors offer guarantees for this solution under several sets of assumptions on the functions.

Cons:
- The theoretical results can be cleaned to offer better guarantees and make them more interpretable. I think the regret bounds offered in Theorem 3.2, 3.3, 4.2, and 4.3 are difficult to parse, but they can be improved. For example, there shouldn't be a dependence on the inverse of the Lipschitz parameters L_0 and L_1 (it doesn't make sense to get a worse bound when the functions are smoother). The dependence on the inverse arises because the chosen step sizes  go to infinity as the Lipschitz constants go to zero. With a better choice of step sizes the regret bounds would be better.

- Assumptions 3.1 and 4.1 are stated in terms of expectations, but it is not clear what the expectations are over. From the proofs it seems that the expectations are over the perturbation directions u, but these are not introduced in the assumptions. Also, is Assumption 4.1.1 really intended as is? I'm asking because it reduces to E f_T <= W_T + E f_0.

- Finally, it seems like in the LQR example the different functions f_t correspond to different transition parameters (A_t, B_t). How are the parameters A_t, B_t chosen? I think a clear discussion of the choice is important to both understand the difficulty of the problem and to understand whether Assumptions 3.1 and 4.1 hold.


-----
Update after rebuttal:

I appreciate the detailed answers to my questions and the authors' revisions. I also read the other reviewers' comments. While the new assumptions address my initial concerns, the new versions depend on the algorithm being implemented. As far as I can tell the assumptions might be satisfied for one choice of step-size while not being satisfied for another. Also, I agree with the other reviewers that generally in OCO one considers a worst case sequence of functions. A discussion of this issue in the main body of the paper seems appropriate.

After addressing these issues, I think this work would warrant acceptance to ICLR. For now, however, I am not changing my score.

---

> ### Author Response · Authors · 2020-11-20
> **Response to reviewer 4**
>
> We thank the reviewer for providing valuable feedback that helps us improve the quality of this paper. Below is a point-to-point response to the reviewer’s concerns.
>
> Q1: Dependence on the inverse of the Lipschitz parameters in main results. With a better choice of step sizes the regret bounds would be better.
>
> A: Thanks for pointing out this. In general, one can choose a different step size $\eta$ and parameter $\delta$ to change the dependence of the complexity on the Lipschitz parameters $L_0, L_1$. However, the best complexity bound will depend on the values of $L_0, L_1$, as we explain below.
>
> First, we want to clarify that the step size choices we make already result in the best dependence of the complexity on the parameters $L_0$ and $L_1$. To elaborate, take Theorem 3.2 as an example. Note that the right hand side of eq. (24) involves the terms $O(\eta^{-1})$ and $O(\eta L_0^2)$, which motivate our optimized step size choice $\eta=O(L_0^{-1})$. Then, we can remove the dependence of the complexity bound on $L_0^{-1}$ by properly choosing $\delta$. Again, take Theorem 3.2 as an example. For the last two terms in eq. (24), we can choose $\delta = O({L_0}^{-1})$ and obtain the final complexity bound $O((L_0 + L_0V_f^2) T^{3/4})$. This bound matches the desired intuition pointed out by the reviewer. However, if the problem has $L_0 > 1$, this new bound will be worse than our current bound in Theorem 3.2.
>
> In Remark 3.3 after Theorem 3.2, we optimize the dependence of the bound on the Lipschitz parameter by choosing $\delta=O(1/L_0^q)$ with $q$ being fine-tuned accordingly to the value of the Lipschitz parameter. The approach in this discussion  can also be applied to optimize the dependence of the complexity bounds in the other Theorems.
>
> Q2: Clarification on Assumption 3.1 and 4.1.
>
> A: We apologize for the confusion. We agree that it is better and more clear to just assume the conditions that are used in the proof. To clarify, we have updated the conditions of these assumptions in the revision.
>
> To elaborate on the randomness, in the revised Assumption 3.1, the expectation is taken with respect to $x_{t-1}$, the random vector $u_{t-1}$ and the randomness of the objective functions $f_t, f_{t-1}$. In the revised Assumption 4.1.1, the expectation in the summation is taken with respect to $x_t$ and the randomness of the smoothed objective functions $f_{\delta, t}, f_{\delta, t-1}$. In the revised Assumption 4.1.2, the expectation in the summation is taken with respect to $x_{t-1}$, the random vector $u_{t-1}$ and the randomness of the objective functions $f_{t}, f_{t-1}$. The Assumptions 5.1 and 5.3 are updated in a similar way, please refer to the revision for the details.
>
> Q3: Clarification on numerical examples.
>
> A: The construction of the matrices $A_t, B_t$ is specified in section A of the supplementary material. Specifically, $A_t$ is generated according to $A_t = A_{t-1} + 0.01*M_t$, where $M_t$ is a random matrix with entries being uniformly sampled from $[0,1]$. Matrix $B_t$ is generated in a similar way. We have clarified these details in Section 6.1.

---

### Decision · Program_Chairs · 2021-01-07
**Final Decision**

**Decision:**

Reject

**Comment:**

The paper generated a lot of discussion. After reviewing all of the opinions, and my own reading of the paper, we have concluded that the theoretical innovation is too incremental for ICLR. It is possible that the idea of "residual feedback" could be helpful, but for this to be demonstrated effectively one would need to consider concrete models where the assumptions are verified.